# Self-Guided Explanation for Graph Neural Networks with Semi-Supervision

## Abstract

Post-hoc explanation for graph neural networks (GNNs) is the task of explaining their decisions by identifying important subgraphs. Since discretization is non-differentiable, most prior work models explainers that output continuous edge-importance scores, often yielding blurry, mixed score distributions. This stems from optimizing only to preserve the original prediction without effective regularization and in the absence of edge-level ground-truth labels. We present 3SG-Explainer (Semi-Supervised and Self-Guided Explainer), which converts weak prediction-preserving signals into explicit edge supervision and, in turn, markedly improves explanation accuracy while sharply polarizing edges into important versus unimportant. Concretely, we introduce confidence-based thresholds to convert noisy soft scores into semi-supervised pseudo-labels, then train a lightweight message-passing explainer on these labels. We also prove that the improved shapes of the score distributions produced by 3SG-Explainer hold against unsupervised baselines. Experiments on four benchmarks and multiple metrics show that 3SG-Explainer improves the accuracy for edge-level explanation over state-of-the-art baselines. To ensure reproducibility, our codes are provided in the supplementary material.

## 1 Introduction

Graph neural networks (GNNs) have emerged as a powerful model for learning over graph-structured data (Kipf and Welling, 2016; Veličković et al., 2017; Xu et al., 2018; Wu et al., 2020), achieving remarkable success across a wide range of domains (Gilmer et al., 2017; Zitnik et al., 2018). Despite the rapid progress on improving the architecture of GNNs, their reasoning process remains unclear, hindering their deployment in settings where trust and accountability are essential (Yuan et al., 2022; Zhang et al., 2024a; Agarwal et al., 2023; Miao et al., 2022).

The *post-hoc explanation* of GNNs therefore seeks to reveal substructures such as nodes, edges, or motifs that drive a model's prediction. Early models such as GNNExplainer (Ying et al., 2019) and PGExplainer (Luo et al., 2020) cast explanation as sampling a sparse edge mask that maximizes the mutual information with the original prediction, while recent approaches leverage causal graphs (Wang et al., 2022), counterfactual explanations (Lucic et al., 2022) or generative modeling (Chen et al., 2024a; Lin et al., 2022) to provide more expressive explanations.

In practice, these post-hoc explainers are trained under indirect supervision without access to ground-truth edge-level annotations, solely by the prediction of the GNN. Most existing methods (Yuan et al., 2021; Ying et al., 2019; Luo et al., 2020; Wang et al., 2021; Xie et al., 2022; Zhang et al., 2023) adopt an information bottleneck objective that aims to identify a sparse subgraph which preserves the original prediction when passed through the pre-trained GNN. Typically, the explainer assigns a single continuous score to each edge to indicate its importance. To facilitate training, a mask entropy regularization term is introduced to encourage the scores to be pushed toward 0 or 1.

However, even when such outputs achieve high AUC, the resulting importance scores often fail to exhibit an intuitive binarization. As illustrated in Figure 1a, the accuracy may appear satisfactory, yet the actual score distribution makes it difficult to identify which edges the model considers truly important or not. What matters most is *human-interpretability:* Our essential objective in applying GNN explainers is not to retain the raw continuous mask values, but to extract the binarized subgraph obtained after thresholding them. However, existing methods provide no clear criterion for where the threshold should be placed, which can result in important edges being excluded from the subgraph or,

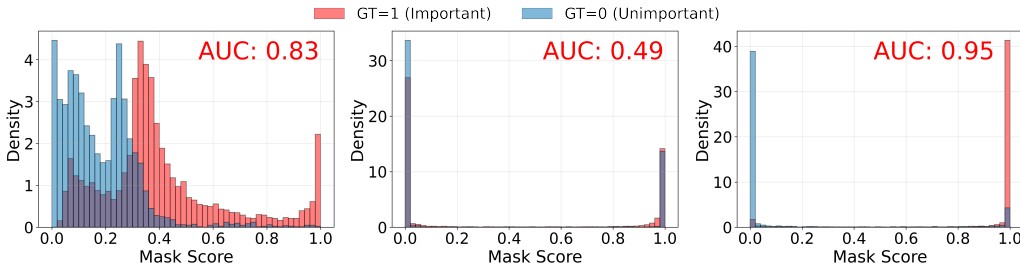

(a) Distribution by PGExplainer.    (b) Over-confident distribution.    (c) Distribution by 3SG-Explainer.

Figure 1: Edge score distributions colored by ground-truth edge labels. (a) A blurry, mixed distribution created by PGExplainer still achieving a high AUC on the BA3 dataset. (b) An over-confident distribution when the entropy term is added and excessively emphasized in PGExplainer for binarization. (c) A desirable distribution produced by 3SG-Explainer, exhibiting both clear polarity and high accuracy.

conversely, unimportant edges being erroneously included. As further illustrated in Figure 1b, a naive reinforcement of entropy regularization is insufficient to effectively control the score distribution, resulting in over-confident predictions. Consequently, without a score distribution that exhibits strong bimodality and clear binarization, it is difficult to regard an explainer as practical or reliable.

As a solution, we propose 3SG-Explainer (Semi-Supervised and Self-Guided Explainer), which is a general framework for enhancing explanation quality through self-guidance from existing base explainers. Our framework does not require modifying or retraining an existing explainer; it aims to extract high-confidence signals and use them to train a guided explainer with semi-supervision. Unlike previous methods that rely solely on the mutual information-based regularization, 3SG-Explainer effectively propagates pseudo-labels derived from the base explainer to the entire graph, providing clear guidance to the guided explainer. This approach results in a clear, binarized score distribution that separates important and unimportant edges even without complex regularization.

Our experiments on four benchmark datasets show that 3SG-Explainer consistently outperforms strong post-hoc explainers with both AUC and F1 scores. Beyond accuracy, 3SG-Explainer yields markedly more interpretable score shapes: binarization and bimodality scores increase across datasets, indicating clearer extraction of the important edges. Ablations show that these gains are robust to the choice of base explainer and persist under round-wise self-guidance. Notably, comprehensive improvements remain strong on real-world molecular graphs and synthetic graphs.

## 2 PROBLEM DEFINITION AND RELATED WORK

**Graph classification.** Each graph is represented as $G = (\mathbf{X}, \mathbf{A})$, where $\mathbf{X} \in \mathbb{R}^{|\mathcal{V}| \times d}$ is the node feature matrix, $\mathbf{A} \in \{0, 1\}^{|\mathcal{V}| \times |\mathcal{V}|}$ is the adjacency matrix, $\mathcal{V}$ is the set of nodes, and $d$ is the feature dimension. We assume undirected graphs. Each element of $\mathbf{A}$ is set to 1 if the corresponding node pair $(u, v)$ is connected, and 0 otherwise. Accordingly, the edge set is defined as $\mathcal{E} \subseteq \mathcal{V} \times \mathcal{V}$, where $(u, v) \in \mathcal{E}$ if and only if $a_{uv} = 1$. In the *graph classification* setting, each graph $G$ is associated with a label $Y \in \{1, \ldots, C\}$ that describes its global structural or semantic property. Given a set $\mathcal{G}$ of training graphs, we assume to have a GNN classifier $f$ trained on $\mathcal{G}$ which is our main target of explanation. All our notations are summarized in Appendix A.

**Post-hoc explanation of GNNs.** Given a pre-trained GNN $f$, a *post-hoc explanation* is to produce a subgraph $G^* \subseteq G$ that most significantly contributes to the GNN's prediction output $\hat{Y} = f(G)$. As the problem is unsupervised, the evaluation of the generated subgraph is typically done by comparing it with the ground-truth edge labels $\mathbf{S} \in \{0, 1\}^{|\mathcal{V}| \times |\mathcal{V}|}$ which encode the importance of each edge for the label $Y$ of graph $G$. Such ground-truth is only for evaluation and not given at training time.

Although the goal of explanation is to find a discrete subgraph, most existing explainer models are designed to return continuous edge importances $\hat{\mathbf{S}} \in [0, 1]^{|\mathcal{V}| \times |\mathcal{V}|}$. We denote an explainer by $h$.

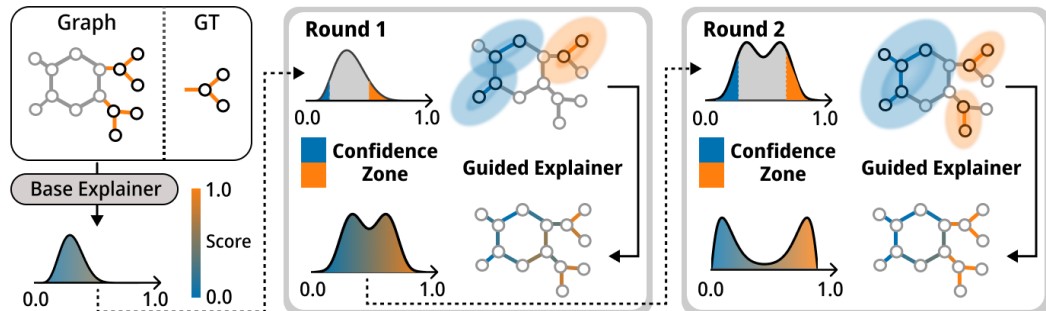

Figure 2: Overall workflow of proposed 3SG-Explainer. Edges falling within the confidence zones (orange or blue regions) are converted into pseudo-labels. The information from these pseudo-labeled edges is then propagated across the graph to provide supervision for unlabeled edges. After one round, pseudo-labeling is repeated, and the guided explainer is fine-tuned with more reliable supervision. Then the score distribution becomes more interpretable, showing clear bimodality and polarization.

**Graph information bottleneck.**    With the graph information bottleneck (GIB) principle, the ideal explanation of GNN $f$ satisfies $\max_{G^*} \text{MI}(\hat{Y}, G^*) = \max_{G^*} \left( H(\hat{Y}) - H(\hat{Y} \mid G^*) \right)$, where $H$ is the entropy function. Since $H(\hat{Y})$ is fixed by the given GNN $f$, maximizing the mutual information is equivalent to minimizing the conditional entropy $H(\hat{Y} \mid G^*)$, which makes the extracted subgraph $G^*$ have the same class information as in the original graph $G$. To enforce compactness of explanation with sparse $G^*$, prior works typically include a regularizer term on its size. This constraint encourages the explainer to focus on a minimal set of salient edges.

**Related work.**    The principle of explanation through mutual information has been consistently adopted across a wide spectrum of works, from early graph mask learning in GNNExplainer (Ying et al., 2019) to parametric prediction with PGExplainer (Luo et al., 2020) and fine-tuning strategies in ReFine (Wang et al., 2021). To address distribution shifts and improve generalization, later GIB-based methods such as MixupExplainer (Zhang et al., 2023) and ProxyExplainer (Chen et al., 2024b) introduced graph mixing and generation. More recently, there have also been explorations into self-explainable temporal models (Seo et al., 2024), factorized explainers (Huang et al., 2024), and regression-based variants (Zhang et al., 2024b). These developments highlight the enduring influence of the GIB-based paradigm in shaping the field of GNN explanation.

## 3 PROPOSED METHOD: 3SG-EXPLAINER

We introduce 3SG-Explainer, which leverages semi-supervised learning and self-guidance to convert weak unsupervised signals into explicit edge-level supervision. As depicted in Figure 2, the process begins with pseudo-label assignment from soft scores, followed by propagation across the graph to exploit structural information. These pseudo-labels guide a lightweight explainer that is retrained in a semi-supervised manner, where only confident edges provide explicit supervision. By iteratively updating labels and finetuning over rounds, the score distribution becomes increasingly polarized and bimodal, yielding clearer separation between important and unimportant edges.

### 3.1 GENERATING PSEUDO-LABELS VIA CONFIDENCE ZONE

3SG-Explainer begins by obtaining mask scores from the base explainer $h^{(b)}$, where each edge in the graph is assigned a continuous value. We aim to assign pseudo-labels using these mask scores, but these scores may be inaccurate depending on the performance of the base explainer. To acquire confident pseudo-labels for self-guidance, we divide the scores into *confidence zone* and *gray zone* based on the thresholds statistically earned from the distributions of scores. Pseudo-labels are then assigned only to the confidence zone, either a positive or a negative label, leaving the gray zone.

The interval between the two thresholds defines the gray zone. Edges with scores outside this interval belong to the confidence zone, which is further divided into a positive and a negative zone. This process yields a new dataset in which only edges with high-confident predictions are equipped with

pseudo-labels derived from the base explainer. To prevent the guided explainer from becoming over-confident with respect to potentially noisy labels, edges in the gray zone are explicitly ignored when calculating the loss. These labeled edges provide reliable supervision for training the guided explainer, while the unlabeled edges remain unused at this stage.

Deciding the thresholds is an important part of our pseudo-labeling. Given the score distribution on the whole dataset from the base explainer, i.e., $P_s = \{\{ s \mid s \in h^{(b)}(G), \ G \in \mathcal{G} \}\}$, where $\{\{\cdot\}\}$ denotes a multi-set. Here, $h^{(b)}(G)$ outputs a continuous score matrix, and all its individual entries $s$ collectively form the distribution. We then compute the skewness $g_1$ (i.e., Fisher's moment coefficient of skewness) (Kendall and Stuart, 1977; Joanes and Gill, 1998) as follows:

$$g_1 = \mathbb{E}_{s \sim P_s}\big[(s - \mu)^3\big]/\sigma^3, \tag{1}$$

where $\mu = \mathbb{E}[s]$ and $\sigma^2 = \mathbb{E}[(s - \mu)^2]$. A positive $g_1$ indicates a longer right tail, while a negative $g_1$ indicates a longer left tail. We then determine asymmetric quantile levels to minimize noise and improve label quality, using a base cutoff $\alpha_0 \in (0, 0.5)$ and a skewness adjustment factor $c > 0$:

$$\alpha_{\text{pos}} = \alpha_0\big(1 + c \cdot \max(g_1, 0)\big), \qquad \alpha_{\text{neg}} = \alpha_0\big(1 + c \cdot \max(-g_1, 0)\big). \tag{2}$$

Finally, the lower and upper thresholds are defined as follows:

$$t_L = Q_{\alpha_{\text{neg}}}(P_s), \qquad t_U = Q_{1-\alpha_{\text{pos}}}(P_s), \tag{3}$$

where $Q_\beta(P_s)$ denotes the $\beta$-th quantile of the distribution of $P_s$. In our pseudo-labeling rule, edges with scores below the lower threshold $t_L$ are assigned label 0, those above the upper threshold $t_U$ are assigned label 1, and edges with scores in between remain unlabeled. This thresholding strategy effectively converts the soft importance scores into discrete supervisory signals, enabling the base explainer to guide the subsequent explainer under semi-supervision.

One may alternatively determine thresholds using fixed symmetric quantiles or a simple mean-based rule. However, both approaches overlook potential imbalance in the score distribution and are prone to noisy labels: the quantile method may place cutoffs in noisy mid-ranges when the distribution is skewed, while the mean-based rule implicitly assumes Gaussian-like symmetry that rarely holds in practice. In contrast, our skewness-adjusted thresholds adapt to the actual tail shape, avoiding ambiguous mid-regions and producing cleaner pseudo-labels.

### 3.2 SEMI-SUPERVISED TRAINING OF THE GUIDED EXPLAINER

As a result of pseudo-labeling, we have a new dataset $\mathcal{G}' = \{(G, \mathbf{S}', \mathbf{M})\}$, where $\mathbf{S}' \in \{0, 1\}^{|\mathcal{V}| \times |\mathcal{V}|}$ represents the edge-wise pseudo-labels and $\mathbf{M} \in \{0, 1\}^{|\mathcal{V}| \times |\mathcal{V}|}$ represents the edge supervision mask; $m_{uv} = 1$ means that the pseudo-label is given for edge $(u, v)$. Our goal is to use the pseudo-labels as stable guidance for training a guided explainer $h^{(g)}$, propagating high-confidence signals to all other edges and addressing the limitations of GIB-based models (detailed analysis in Section 3.3).

**Model architecture.** For effective semi-supervised learning, we adopt a GNN architecture in $h^{(g)}$ instead of the MLP backbone used in base explainer $h^{(b)}$ used in our experiments (Luo et al., 2020). The base explainer $h^{(b)}$ takes the node embeddings $\mathbf{Z}$ generated by the GNN $f$ as its input, which is the main reason why it uses the simple MLP architecture. On the other hand, $h^{(g)}$ does not rely on $\mathbf{Z}$. GNN message passing propagates the signal to neighboring edges through repeated aggregation, aligning unlabeled embeddings with the labeled edges even with insufficient supervision.

By default, we use a two-layer GCN followed by an edge classifier as $h^{(g)}$. Given $G = (\mathbf{X}, \mathbf{A})$, the node embeddings $\mathbf{H}$ are computed as $\mathbf{H} = \sigma(\hat{\mathbf{A}}\,\sigma(\hat{\mathbf{A}}\mathbf{X}\mathbf{W}_1)\mathbf{W}_2)$, where $\hat{\mathbf{A}} \in \mathbb{R}^{|\mathcal{V}| \times |\mathcal{V}|}$ is the symmetrically normalized adjacency matrix, $\mathbf{W}_1$ and $\mathbf{W}_2$ are learnable weights, and $\sigma(\cdot)$ is the ReLU function. For each edge $(u, v)$ in the graph $G'$, we extract the concatenated node embeddings and pass it through a linear edge classifier to obtain the final edge importance:

$$\hat{s}_{uv}^{(g)} = \sigma([\mathbf{h}_u \| \mathbf{h}_v]\mathbf{W}_3 + \mathbf{b}), \tag{4}$$

where $\mathbf{W}_3$ and $\mathbf{b}$ are learnable parameters.

**Training.** The cross-entropy loss is typically used for binary classification tasks. However, in our case, only a small fraction of edges can be considered important, especially in certain domains like

molecular graphs. The overwhelming majority are easy negatives whose presence scarcely affects the graph label. To address this imbalance, we adopt a *positive class-weighted* binary cross-entropy (BCE) loss, so that positive edges contribute more strongly during training:

$$\mathcal{L}_g(h^{(g)}) = \sum_{(u,v)\in\mathcal{E}} \left[ w^+ s'_{uv} \log \hat{s}^{(g)}_{uv} + (1 - s'_{uv}) \log\big(1 - \hat{s}^{(g)}_{uv}\big) \right], \tag{5}$$

where $\mathcal{E}$ is the set of edges in $G$, $s'_{uv} \in \{0, 1\}$ is the pseudo-label for edge $(u, v)$, and $w^+ > 1$ is the weight assigned to positive edges to compensate for their scarcity.

**Self-guided fine-tuning.** We further enhance the guided explainer by introducing a *self-guidance* strategy that proceeds for additional rounds. In the second round, we regenerate pseudo-labels from the guided explainer $h^{(g)}$, not the base explainer $h^{(b)}$, and fine-tune the same guided explainer as done in the first round. This multi-round design ensures that the explainer adapts to the newly generated pseudo-labels while retaining useful representations learned in the earlier stage, gradually improving the score distribution. Since later rounds tend to provide higher-quality supervision, with more (pseudo-labeled) edges in the confidence zone, fine-tuning allows the model to consolidate the refined signals without discarding previously learned discriminative features. We later show in an experiment (in Figure 4) that the quality of score distribution gradually improves as the round progresses.

### 3.3 THEORETICAL INSIGHTS

We provide theoretical insights on the reason why existing Graph information bottleneck (GIB)-based methods fail at generating binarized edge score distributions, while 3SG-Explainer can.

**Why GIB fails at binarization.** The binarization of edge important scores may be achieved by adding a suitable regularizer to the GIB objective function, not only by our self-guided framework. Previous work (Luo et al., 2020) proposes an objective function that balances prediction consistency and the sparsity and binarization of edge importance scores as follows:

$$\mathcal{L}_b(h^{(b)}) = \text{CE}\big(\hat{Y}, \hat{Y}^*\big) + \lambda_1 \sum_{(u,v)\in\mathcal{E}} \hat{s}^{(b)}_{uv} + \lambda_2 \sum_{(u,v)\in\mathcal{E}} H(\hat{s}^{(b)}_{uv}), \tag{6}$$

where $H$ is the entropy function that is minimized if the scalar input is either 0 or 1, $Y^* = f(G^*)$ is the prediction of the GNN $f$ for the extracted subgraph $G^* = (\mathbf{X}, \hat{\mathbf{S}}^{(b)})$, and $\hat{s}^{(b)}_{uv} \in [0, 1]$ is the soft importance score for edge $(u, v) \in \mathcal{E}$ produced by the explainer $h^{(b)}$. The first term measures consistency with the original GNN prediction, the second term encourages score sparsity, and the third term promotes binarization, where the coefficients $\lambda_1$ and $\lambda_2$ make a balance.

The main limitation of Eq. 6 is that it has a provable tendency to converge to non-binary scores in the middle of 0 and 1, not being polarized sufficiently, as formalized below.

**Lemma 1** (Failure of binarization for GIB). *For the regularized GIB loss $\mathcal{L}_b$ in Eq. (6), consider a stationary point where the fidelity gradient $g_{uv}(\hat{\mathbf{S}}^{(b)}) := \partial \text{CE}(\hat{Y}, \hat{Y}^*)/\partial\hat{s}^{(b)}_{uv}$ is zero. At this point, the optimal score $\hat{s}^{(b)}_{uv}$ converges to a fractional value independent of the edge properties:*

$$\hat{s}^{(b)\star}_{uv} = \sigma(\lambda_1/\lambda_2) \in (0, 1), \tag{7}$$

*where $\sigma$ is the logistic sigmoid function.*

The detailed derivation and the proof of Lemma 1 are provided in Appendix B.1. The lemma states that unlike our expectation, there can be an upper bound of binarization to achieve with GIB.

**Why 3SG-Explainer succeeds in binarization.** Unlike the GIB objective function, our semi-supervised training of the guided explainer can result in clearer binarization. Let $z_{uv}$ be the score logit for edge $(u, v)$, which is a real-valued version of $\hat{s}^{(g)}_{uv} = \sigma(z_{uv})$ before being applied to the logistic sigmoid function $\sigma$. Given the BCE objective function in Eq. 5, we first provide a lemma that the gradient of the loss function pushes logits away from the decision boundary on the confident tail (i.e., confidence zone). The formal proof is provided in Appendix B.2.

**Lemma 2** (Binarization in the confidence zone). *Let the BCE loss gradient be $\partial\ell/\partial z_{uv} = \hat{s}^{(g)}_{uv} - s'_{uv}$. If the tail noise rates satisfy $\varepsilon_\pm < 1/2$, the expected gradient pushes scores to their pseudo-labels. Conditioned on the upper tail $L_+$, this expectation is negative ($\mathbb{E}[\cdot|L_+] < 0$), causing $\hat{s}^{(g)}_{uv} \to 1$. Conversely, on the lower tail $L_-$, it is positive ($\mathbb{E}[\cdot|L_-] > 0$), causing $\hat{s}^{(g)}_{uv} \to 0$.*

While Lemma 2 explains the direct binarization of pseudo-labeled edges, the GNN architecture of the guided explainer propagates these signals to unlabeled edges in the gray zone. This propagation via shared weights and message passing is formalized below.

**Corollary 1** (Binarization in the gray zone). *Let $(u, v)$ be an unlabeled edge. The change in its score, $\Delta \hat{s}_{uv}^{(g)}$, after one gradient step with learning rate $\eta$, is determined by the alignment of its gradient with the gradients of labeled edges in its structural neighborhood. Formally,*

$$\Delta \hat{s}_{uv}^{(g)} \approx -\eta \sum_{(i,j)\in L} \frac{\partial l_w}{\partial \hat{s}_{ij}^{(g)}} \langle \nabla_\theta \hat{s}_{uv}^{(g)}, \nabla_\theta \hat{s}_{ij}^{(g)} \rangle,$$

*where the inner product is large and positive for structurally proximate edges. Consequently, the score of an unlabeled edge is pushed toward the dominant pseudo-label of its labeled neighbors.*

Lemma 2 and Corollary 1 explain how 3SG-Explainer achieves strong binarization. On confident edges, scores are directly pushed toward 0 or 1. These signals are then propagated through the GNN explainer to unlabeled edges, pushing their scores toward the consensus of their local neighborhood. Consequently, (i) scores move away from the decision boundary, $0.5$, across the entire graph; (ii) re-scoring edges after each training stage shifts more edges into the tails and fewer remain in the gray zone; and (iii) repeating the self-guided loop (relabeling $\rightarrow$ retrain) through rounds amplifies this effect, making the edge-score distribution increasingly binary over time.

## 4 EXPERIMENTS

We conduct comprehensive experiments on benchmarks to verify the performance of 3SG-Explainer.

### 4.1 EXPERIMENTAL SETTINGS

**Datasets.** We run experiments on two types of graph datasets: synthetic graphs and real-world graphs. For synthetic graphs, we use the BA-3motifs dataset (Wang et al., 2021) and MNIST-75sp dataset (Monti et al., 2017). For real-world data, we include three molecular datasets, Mutagenicity (Debnath et al., 1991), Fluoride-Carbonyl (Sanchez-Lengeling et al., 2020), and Benzene (Sanchez-Lengeling et al., 2020). Since graph classification datasets with edge-level ground-truth are very scarce, we carefully select these benchmarks to cover both synthetic and real-world domains. The details of the datasets and the corresponding GNN $f$ used for each dataset are provided in Appendix C.

**Baselines.** We compare 3SG-Explainer with strong explainer methods, including PGExplainer (Luo et al., 2020), ReFine (Wang et al., 2021), TAGE (Xie et al., 2022), MixupExplainer (Zhang et al., 2023) and ProxyExplainer (Chen et al., 2024b). These baselines are post-hoc explainers built upon the GIB principle, aiming to generate subgraphs that preserve the original GNN prediction. While grounded in the same objective, each method tackles a different subproblem such as refinement of coarse masks, task-agnostic generalization, or distribution alignment to improve the interpretability of the explanations (Liu et al., 2021). In addition to these GIB-based methods, we also include non-GIB, training-free approaches such as EiG-Search (Lu et al., 2024a) and GOAt (Lu et al., 2024b), which provide analytic and architecture-agnostic explanations without learning auxiliary models. Further details of each baseline are provided in Appendix D.

**Evaluation metrics.** We present a comprehensive set of metrics that jointly capture predictive correctness and distributional interpretability. For predictive correctness, we report AUC and F1 scores in comparison with existing baselines. For clarity of the score distribution, we introduce bimodality and binarization scores, which directly reflect our motivation of shaping edge-score distributions into a more human-interpretable form. bimodality represents whether the distribution exhibits two distinct modes, and binarization score represents the degree to which scores concentrate near 0 or 1. To the best of our knowledge, these distributional metrics are the first to be proposed in the context of GNN explainers, highlighting how our method goes beyond accuracy to evaluate explanation quality. Detailed equations related to distribution are provided in Appendix E.

**Implementation details.** We adopt PGExplainer (Luo et al., 2020) as the base explainer of 3SG-Explainer, owing to its model-level parameterization, fast training speed, and strong AUC performance. To ensure robustness, we later verify that our approach yields consistent improvements when combined with alternative base explainers. For evaluation, we report the results from the second

round of 3SG-Explainer for both AUC and F1 score. Each experiment is repeated over 10 random seeds, and the mean and standard deviation are reported. Depending on the dataset, we use either a 7:1.5:1.5 split or a 6:2:2 split for training, validation, and testing. The main hyperparameters are $\alpha_0$, $c$, and $w^+$, which are tuned via grid search. Further implementation details are provided in Appendix F.

## 4.2 PERFORMANCE IMPROVEMENT

Table 1: Explanation correctness in terms of AUC-ROC scores. The best model in each dataset is highlighted in bold, while the runner-up is underlined.

| Method | BA-3motifs | Mutagenicity | MNIST-75sp | Fluoride-Carbonyl | Benzene |
|---|---|---|---|---|---|
| PGExplainer (2020) | $0.780 \pm 0.070$ | $0.801 \pm 0.026$ | $0.732 \pm 0.012$ | $0.749 \pm 0.042$ | $0.695 \pm 0.002$ |
| Refine (2021) | $0.823 \pm 0.050$ | $0.534 \pm 0.099$ | $0.623 \pm 0.019$ | $0.754 \pm 0.032$ | $0.762 \pm 0.187$ |
| TAGE (2022) | $0.709 \pm 0.084$ | $0.756 \pm 0.087$ | $0.714 \pm 0.028$ | $0.779 \pm 0.054$ | $0.354 \pm 0.159$ |
| MixupExplainer (2023) | $0.732 \pm 0.085$ | $0.822 \pm 0.057$ | $0.727 \pm 0.003$ | $0.729 \pm 0.075$ | $0.483 \pm 0.008$ |
| ProxyExplainer (2024) | $0.742 \pm 0.073$ | $0.805 \pm 0.027$ | $0.732 \pm 0.013$ | $0.754 \pm 0.037$ | $0.465 \pm 0.361$ |
| Eig-Search (2024) | $0.624 \pm 0.000$ | $0.487 \pm 0.000$ | $0.511 \pm 0.000$ | $0.779 \pm 0.000$ | $0.910 \pm 0.000$ |
| GOAt (2024) | $0.770 \pm 0.000$ | $0.663 \pm 0.000$ | $0.563 \pm 0.000$ | $0.780 \pm 0.000$ | $0.719 \pm 0.000$ |
| 3SG-Explainer | $\mathbf{0.906 \pm 0.030}$ | $\mathbf{0.916 \pm 0.029}$ | $\mathbf{0.914 \pm 0.010}$ | $\mathbf{0.822 \pm 0.047}$ | $\mathbf{0.912 \pm 0.007}$ |

Table 2: F1 scores evaluated at the median threshold across datasets. The best model in each dataset is highlighted in bold, while the runner-up is underlined.

| Method | BA-3motifs | Mutagenicity | MNIST-75sp | Fluoride-Carbonyl | Benzene |
|---|---|---|---|---|---|
| PGExplainer (2020) | $0.553 \pm 0.207$ | $0.259 \pm 0.003$ | $0.503 \pm 0.012$ | $0.403 \pm 0.095$ | $0.647 \pm 0.006$ |
| Refine (2021) | $0.788 \pm 0.004$ | $0.197 \pm 0.042$ | $0.425 \pm 0.013$ | $0.354 \pm 0.039$ | $0.372 \pm 0.225$ |
| TAGE (2022) | $0.657 \pm 0.100$ | $0.265 \pm 0.099$ | $0.492 \pm 0.022$ | $0.455 \pm 0.108$ | $0.361 \pm 0.165$ |
| MixupExplainer (2023) | $0.699 \pm 0.054$ | $0.298 \pm 0.011$ | $0.499 \pm 0.002$ | $0.429 \pm 0.063$ | $0.524 \pm 0.002$ |
| ProxyExplainer (2024) | $0.717 \pm 0.043$ | $0.297 \pm 0.004$ | $0.505 \pm 0.012$ | $0.458 \pm 0.026$ | $0.384 \pm 0.327$ |
| EiG-Search (2024) | $0.736 \pm 0.000$ | $0.145 \pm 0.000$ | $0.370 \pm 0.000$ | $0.465 \pm 0.000$ | $0.763 \pm 0.000$ |
| GOAt (2024) | $0.744 \pm 0.000$ | $0.214 \pm 0.000$ | $0.385 \pm 0.000$ | $0.483 \pm 0.000$ | $0.594 \pm 0.000$ |
| 3SG-Explainer | $\mathbf{0.802 \pm 0.023}$ | $\mathbf{0.324 \pm 0.003}$ | $\mathbf{0.661 \pm 0.008}$ | $\mathbf{0.514 \pm 0.043}$ | $\mathbf{0.797 \pm 0.001}$ |

Table 1 compares the explanation accuracy of 3SG-Explainer with the baselines. 3SG-Explainer consistently outperforms the baselines by a significant margin across all datasets, showing particularly large gains in AUC compared to the best-performing single-stage methods. The gains remain strong on the real-world benchmarks (Mutagenicity, Fluoride-Carbonyl, Benzene), indicating that 3SG-Explainer is still effective even when edge labels are scarce and noise is present. Among the datasets, MNIST-75sp is characterized by significantly higher edge density compared to other datasets, presenting a particularly challenging scenario with densely connected structures and noisy edge patterns. The notable performance improvement of 3SG-Explainer on MNIST-75sp demonstrates its robustness and versatility when applied to complex and high-density graphs.

As shown in Table 2, 3SG-Explainer consistently achieves the highest F1 scores across all datasets, outperforming all baselines. Unlike AUC, F1 score requires a decision threshold on the predicted edge scores. F1 score evaluates the explainer's ability to make concrete binary decisions by balancing precision and recall. We adopt the median score as a natural and unbiased threshold, thereby avoiding dataset-specific tuning and providing a fair criterion across different methods. Across all four datasets, 3SG-Explainer outperforms the best competitors by clear margins. These results indicate that semi-supervised learning with self-guidance enables the explainer to better distinguish true positives and true negatives in practical decision-making scenarios than the base explainer.

## 4.3 DISTRIBUTIONAL ANALYSIS AND VISUAL INTERPRETABILITY

We provide qualitative and quantitative analysis for a deeper understanding of 3SG-Explainer. These visualizations are intended (i) to assess the interpretability proposed by the guided explainer and (ii) to evaluate whether the resulting score distributions are sufficiently polarized and bimodal.

Table 3: Evaluation of base and guided explainers across four datasets in terms of binarization score and bimodality. The guided explainer consistently improves both metrics over the base explainer, with the second round achieving the highest scores overall. This indicates that repeated self-guidance yields clearer separation and more strongly bimodal score distributions.

| Method | Metric | BA3 | Mutagenicity | MNIST-75sp | FC | Benzene |
|---|---|---|---|---|---|---|
| Base Explainer (PGExplainer) | Binarization | $0.688 \pm 0.287$ | $0.128 \pm 0.091$ | $0.009 \pm 0.009$ | $0.000 \pm 0.000$ | $0.857 \pm 0.021$ |
| | Bimodality | $0.631 \pm 0.304$ | $0.440 \pm 0.029$ | $0.124 \pm 0.066$ | $0.063 \pm 0.101$ | $0.880 \pm 0.023$ |
| Guided Explainer (Round 1) | Binarization | $0.615 \pm 0.341$ | $0.502 \pm 0.223$ | $0.285 \pm 0.254$ | $0.570 \pm 0.407$ | $0.720 \pm 0.098$ |
| | Bimodality | $0.508 \pm 0.361$ | $0.539 \pm 0.177$ | $0.460 \pm 0.226$ | $0.756 \pm 0.276$ | $0.558 \pm 0.135$ |
| Guided Explainer (Round 2) | **Binarization** | $\mathbf{0.835 \pm 0.157}$ | $\mathbf{0.591 \pm 0.254}$ | $\mathbf{0.666 \pm 0.328}$ | $\mathbf{0.851 \pm 0.245}$ | $\mathbf{0.890 \pm 0.010}$ |
| | **Bimodality** | $\mathbf{0.826 \pm 0.240}$ | $\mathbf{0.670 \pm 0.184}$ | $\mathbf{0.722 \pm 0.232}$ | $\mathbf{0.872 \pm 0.243}$ | $\mathbf{0.896 \pm 0.011}$ |

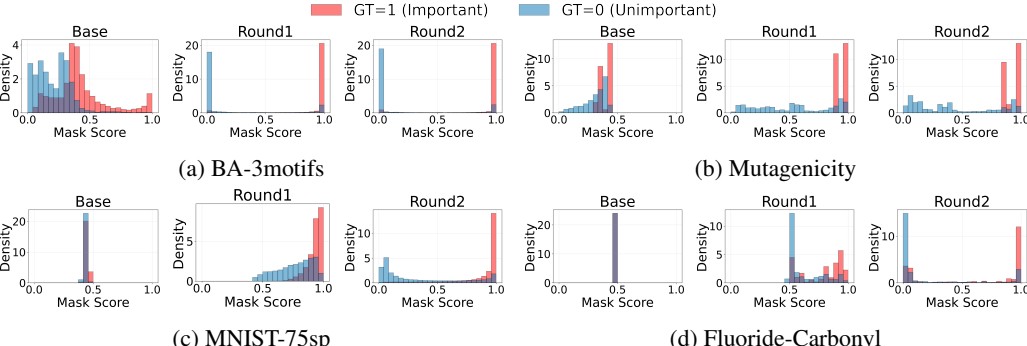

Figure 3: Trends of score distributions created by 3SG-Explainer across four datasets. For each dataset, the three plots correspond to the base explainer, the first round, and the second round. As rounds progress, the distributions become more polarized and bimodal, resulting in clearer separation.

As Table 3 shows, the score distributions produced by the base explainer are unreliable across the datasets, with scores strongly skewed toward a narrow range near specific values, and thus fail to exhibit clear separation between label 0 and label 1 edges. In comparison, 3SG-Explainer alleviates this bias and produces more balanced, polarized, and interpretable distributions with consistently predictive accuracy. Across whole datasets, both binarization and bimodality consistently increase, showing that the semi-supervised and self-guided training drives the scores toward a more binarized and interpretable two-mode structure. Considering the accuracy metrics in Table 1, and 2, these distributional gains are accompanied by improved task performance.

Our primary objective is to construct a score distribution that is human-interpretable. In real-world scenarios, explainer models are deployed without access to ground-truth edge labels. Thus, users must rely solely on the raw score distribution to determine an appropriate threshold for extracting a discrete subgraph. Figure 3 compares the distributions produced by the base explainer and our guided explainer on four datasets. For base explainers, we observe that edges with label 0 and 1 are barely distinguishable because the two distributions overlap in most regions, with the distribution of label 1 edges assigned only slightly higher scores on average. As a result, it remains unclear where thresholding should be applied to get a clear and concise subgraph from the base explainer itself.

In contrast, after applying 3SG-Explainer, the modified distribution becomes binarized toward 0 and 1 rather than clustered within a limited region. With the increase in rounds, the distributions become progressively more polarized and interpretable, and the associated performance improvements become evident. As a result, users can identify a more confident thresholding interval with greater clarity.

## 4.4 ABLATION STUDIES

We perform two types of ablation studies considering 3SG-Explainer as a modular framework: *(i) Base explainer variation.* We replace the base explainer with different alternatives to assess how robust our module remains under varying base components; *(ii) Round-wise training variation.* We

Table 4: Performance improvements of 3SG-Explainer over different base explainers. Note that PG-Explainer is used by default in other experiments. 3SG-Explainer consistently improves explanation quality across base explainers, demonstrating its general applicability.

| Base Explainer | Dataset | AUC | | Binarization score | | Bimodality | |
|---|---|---|---|---|---|---|---|
| | | Base | Guided | Base | Guided | Base | Guided |
| PGExplainer | Mutagenicity | $0.801 \pm 0.026$ | $\mathbf{0.916 \pm 0.030}$ | $0.128 \pm 0.091$ | $\mathbf{0.591 \pm 0.254}$ | $0.440 \pm 0.029$ | $\mathbf{0.670 \pm 0.184}$ |
| | MNIST-75sp | $0.732 \pm 0.012$ | $\mathbf{0.914 \pm 0.010}$ | $0.009 \pm 0.009$ | $\mathbf{0.666 \pm 0.328}$ | $0.124 \pm 0.066$ | $\mathbf{0.722 \pm 0.232}$ |
| MixupExplainer | Mutagenicity | $0.805 \pm 0.027$ | $\mathbf{0.922 \pm 0.033}$ | $0.055 \pm 0.075$ | $\mathbf{0.619 \pm 0.257}$ | $0.460 \pm 0.036$ | $\mathbf{0.680 \pm 0.176}$ |
| | MNIST-75sp | $0.727 \pm 0.003$ | $\mathbf{0.914 \pm 0.006}$ | $0.010 \pm 0.009$ | $\mathbf{0.799 \pm 0.066}$ | $0.130 \pm 0.017$ | $\mathbf{0.816 \pm 0.053}$ |
| ProxyExplainer | Mutagenicity | $0.805 \pm 0.027$ | $\mathbf{0.907 \pm 0.023}$ | $0.132 \pm 0.094$ | $\mathbf{0.675 \pm 0.100}$ | $0.437 \pm 0.030$ | $\mathbf{0.666 \pm 0.081}$ |
| | MNIST-75sp | $0.732 \pm 0.013$ | $\mathbf{0.908 \pm 0.015}$ | $0.010 \pm 0.011$ | $\mathbf{0.775 \pm 0.059}$ | $0.121 \pm 0.018$ | $\mathbf{0.791 \pm 0.049}$ |

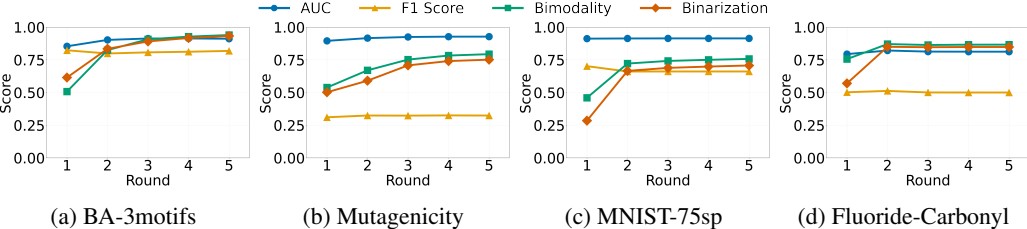

(a) BA-3motifs  (b) Mutagenicity  (c) MNIST-75sp  (d) Fluoride-Carbonyl

Figure 4: Change of various metrics over multiple rounds of 3SG-Explainer. As rounds proceed, AUC tends to increase slightly or remain similar, while distributional metrics like bimodality and binarization score steadily rise and converge. These demonstrate that increasing rounds strengthens the polarization of the score distribution while preserving predictive accuracy.

examine how performance evolves as the number of rounds increases, showing whether our module converges stably and yields progressively more interpretable explanations.

**Base explainer variation.** We evaluate the robustness of 3SG-Explainer by substituting different base explainers. As reported in Table 4, 3SG-Explainer consistently improves AUC as well as distributional metrics such as binarization score and bimodality, independent of whether the underlying base explainer is ProxyExplainer or MixupExplainer. These results indicate that the effectiveness of our module does not depend on a particular explainer architecture.

**Round-wise training variation.** We further analyze how the performance of 3SG-Explainer evolves as the number of rounds increases over 2. In Figure 4, while AUC shows little change beyond slight improvements, binarization score and bimodality continue to rise and eventually converge, revealing progressively sharper distinctions between positive and negative edges. For F1 scores, the highest values typically appear in the initial 1–2 rounds, after which they decrease slightly but eventually stabilize, reflecting a trade-off where sharper and more interpretable score distributions are achieved at the cost of marginal predictive balance. Overall, these results demonstrate that our multi-round self-guidance scheme converges stably and enhances interpretability over successive rounds.

## 5 CONCLUSION

We presented 3SG-Explainer, a post-hoc explainer for graph neural networks designed to drive score distributions toward clearer binarization and bimodality for improved interpretability. 3SG-Explainer transforms soft edge importance scores into reliable supervisory signals and amplifies them through self-guidance, progressively sharpening edge-level distinctions. We further provided theoretical analysis showing why semi-supervision with pseudo-labeling drives scores away from ambiguous mid-ranges, yielding stronger binarization compared to prior GIB-based approaches. Across both synthetic and real-world datasets, 3SG-Explainer consistently outperforms baselines in terms of accuracy, while remaining interpretable and easily integrable. Beyond these results, our framework motivates several future directions, including adaptive round selection or threshold-free approaches to pseudo-labeling.

## ETHICS STATEMENT

Our work focuses on developing explainability methods for graph neural networks using publicly available benchmark datasets. Our experiments do not involve human subjects, sensitive personal information, or any data raising privacy concerns. We anticipate no direct negative societal impacts, and the goal of this research is to advance understanding of trustworthy and interpretable machine learning models. We encourage the responsible use of our methods and align with the ICLR Code of Ethics by upholding fairness, transparency, and integrity in both research and its applications.

## REPRODUCIBILITY STATEMENT

To facilitate reproducibility, we provide our implementation along with datasets used in the experiments. Due to file size constraints, MNIST-75sp must be downloaded from the reference in the paper. The code release includes the best hyperparameter settings for each dataset, while the search ranges are provided in Appendix F. We also attach the pre-trained parameters of the base explainers used in our experiments. Furthermore, the code is configured to support multiple random seeds, allowing readers to easily reproduce and verify our reported results.

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

# A NOTATIONS

Table 5: Summary of notation.

| Symbol | Description |
|---|---|
| $G = (\mathbf{X}, \mathbf{A})$ | Input graph |
| $G^* = (X, A^*)$ | Explanatory subgraph sampled by explainer |
| $\mathbf{X} \in \mathbb{R}^{|\mathcal{V}| \times d}$ | Node feature matrix of dimension $d$ |
| $\mathbf{A} \in \{0, 1\}^{|\mathcal{V}| \times |\mathcal{V}|}$ | Adjacency matrix |
| $f$ | Pre-trained GNN classifier |
| $Y \in \{1, \ldots, C\}$ | Graph-level label with $C$ classes |
| $\hat{Y} = f(G) \in \{1, \ldots, C\}$ | Prediction of the GNN on input graph |
| $\hat{Y}^* = f(G^*) \in \{1, \ldots, C\}$ | Prediction of the GNN on the explanatory subgraph |
| $h^{(b)}$ | Base explainer |
| $h^{(g)}$ | Guided explainer |
| $\mathbf{S} \in \{0, 1\}^{|\mathcal{V}| \times |\mathcal{V}|}$ | Ground-truth edge importance matrix |
| $\mathbf{S}' \in \{0, 1\}^{|\mathcal{V}| \times |\mathcal{V}|}$ | Pseudo-labeled edge set |
| $\mathbf{M} \in \{0, 1\}^{|\mathcal{V}| \times |\mathcal{V}|}$ | Edge supervision mask (1 if pseudo-label is assigned) |
| $\hat{\mathbf{S}}^{(b)} \in [0, 1]^{|\mathcal{V}| \times |\mathcal{V}|}$ | Importance score matrix from base explainer |
| $\hat{\mathbf{S}}^{(g)} \in [0, 1]^{|\mathcal{V}| \times |\mathcal{V}|}$ | Importance score matrix from guided explainer |
| $s_{uv} \in \{0, 1\}$ | Ground-truth edge importance score of edge $(u, v)$ |
| $s'_{uv} \in \{0, 1\}$ | Pseudo-label assigned to edge $(u, v)$ |
| $\hat{s}_{uv}^{(b)} \in [0, 1]$ | Predicted importance score of edge $(u, v)$ by base explainer |
| $\hat{s}_{uv}^{(g)} \in [0, 1]$ | Predicted importance score of edge $(u, v)$ by guided explainer |
| $L_b(h^{(b)})$ | Training loss of base explainer |
| $L_g(h^{(g)})$ | Training loss of guided explainer |

# B THEORETICAL INSIGHTS

## B.1 DERIVATION OF THE GIB STATIONARY POINT

The GIB surrogate in Equation (6) can be differentiated coordinate-wise as

$$\frac{\partial \mathcal{L}_b}{\partial \hat{s}_{uv}^{(b)}} = \underbrace{\frac{\partial}{\partial \hat{s}_{uv}^{(b)}} \operatorname{CE}\big(\hat{Y}, \hat{Y}^*(\hat{\mathbf{S}}^{(b)})\big)}_{g_{uv}(\hat{\mathbf{S}}^{(b)})} + \lambda_1 + \lambda_2 \log \frac{1 - \hat{s}_{uv}^{(b)}}{\hat{s}_{uv}^{(b)}} \tag{8}$$

Here $g_{uv}(\hat{\mathbf{S}}^{(b)}) = \langle \frac{\partial \operatorname{CE}}{\partial z}, \frac{\partial z}{\partial \hat{s}_{uv}^{(b)}} \rangle$ by the chain rule with logit $z$ of $f(G^*)$. When the masked prediction already matches (or nearly matches) the original one, the fidelity gradient satisfies $\|\frac{\partial \operatorname{CE}}{\partial z}\| \approx 0$, and often $\frac{\partial z}{\partial \hat{s}_{uv}^{(b)}} = 0$ for many edges. On $g_{uv} \approx 0$, this stationary condition yields a closed-form solution strictly between 0 and 1. Starting from the gradient condition in Equation equation 8, the stationary point satisfies

$$0 = g_{uv} + \lambda_1 + \lambda_2 \log \frac{1 - \hat{s}_{uv}^{(b)}}{\hat{s}_{uv}^{(b)}}.$$

On $g_{uv} \approx 0$, this yields

$$s_{uv}^{(g)\star} = \frac{1}{1 + \exp(-\lambda_1/\lambda_2)} = \sigma(\lambda_1/\lambda_2) \in (0, 1).$$

That is, each coordinate admits a natural fractional fixed point strictly between 0 and 1.

## B.2 DERIVATION OF EXPECTED PUSH DIRECTION FOR CONFIDENCE ZONE

*Proof of Lemma 2.* We provide a formal proof for the two cases stated in the lemma. The core of the argument is that the pseudo-label $s'_{uv}$ is constant within each tail, which creates a systematic bias in the gradient of the loss function.

**Case 1: Upper Tail ($L_+$).** We want to show that $\mathbb{E}\left[\frac{\partial \ell}{\partial z_{uv}} \mid (u,v) \in L_+\right] < 0$.

By definition, the gradient of the binary cross-entropy (BCE) loss $\ell$ with respect to the edge logit $z_{uv}$ is given by:

$$\frac{\partial \ell}{\partial z_{uv}} = \hat{s}^{(g)}_{uv} - s'_{uv}$$

where $\hat{s}^{(g)}_{uv} = \sigma(z_{uv})$ is the output of the sigmoid function.

We take the conditional expectation over all edges $(u,v)$ belonging to the upper tail $L_+ = \{(u,v) : \hat{s}^{(b)}_{uv} \geq t_U\}$:

$$\mathbb{E}\left[\frac{\partial \ell}{\partial z_{uv}} \mid L_+\right] = \mathbb{E}\left[\hat{s}^{(g)}_{uv} - s'_{uv} \mid L_+\right]$$

By the linearity of expectation, this can be separated into two terms:

$$\mathbb{E}\left[\frac{\partial \ell}{\partial z_{uv}} \mid L_+\right] = \mathbb{E}\left[\hat{s}^{(g)}_{uv} \mid L_+\right] - \mathbb{E}[s'_{uv} \mid L_+]$$

According to the pseudo-labeling rule defined in Section 3.1, any edge $(u,v)$ in the upper confidence zone $L_+$ is assigned the pseudo-label $s'_{uv} = 1$. This value is constant for all members of the set $L_+$. Therefore, its expectation is:

$$\mathbb{E}[s'_{uv} \mid L_+] = \mathbb{E}[1 \mid L_+] = 1$$

Substituting this back, we get:

$$\mathbb{E}\left[\frac{\partial \ell}{\partial z_{uv}} \mid L_+\right] = \mathbb{E}\left[\hat{s}^{(g)}_{uv} \mid L_+\right] - 1$$

Since the score $\hat{s}^{(g)}_{uv}$ is the output of a sigmoid function, it is strictly bounded in the interval $[0,1]$. Its expectation over any set of edges must also lie within this interval. Unless the guided explainer has perfectly converged such that $\hat{s}^{(g)}_{uv} = 1$ for all edges in $L_+$ (a trivial stationary point), the expectation will be strictly less than 1:

$$\mathbb{E}\left[\hat{s}^{(g)}_{uv} \mid L_+\right] < 1$$

It follows that the expected gradient is strictly negative:

$$\mathbb{E}\left[\frac{\partial \ell}{\partial z_{uv}} \mid L_+\right] < 0$$

In a gradient descent update step, the change in the logit is proportional to the negative of the gradient (i.e., $\Delta z_{uv} \propto -\frac{\partial \ell}{\partial z_{uv}}$). Since the expected gradient is negative, the expected change in the logit is positive, causing $z_{uv}$ to increase on average and thus pushing the score $\hat{s}^{(g)}_{uv}$ towards 1.

**Case 2: Lower Tail ($L_-$).** The argument is symmetric for the lower tail $L_- = \{(u,v) : \hat{s}^{(b)}_{uv} \leq t_L\}$. The conditional expectation is:

$$\mathbb{E}\left[\frac{\partial \ell}{\partial z_{uv}} \mid L_-\right] = \mathbb{E}\left[\hat{s}^{(g)}_{uv} \mid L_-\right] - \mathbb{E}[s'_{uv} \mid L_-]$$

For any edge in $L_-$, the assigned pseudo-label is $s'_{uv} = 0$. Thus, $\mathbb{E}[s'_{uv} \mid L_-] = 0$. The expression simplifies to:

$$\mathbb{E}\left[\frac{\partial \ell}{\partial z_{uv}} \mid L_-\right] = \mathbb{E}\left[\hat{s}^{(g)}_{uv} \mid L_-\right]$$

Unless the model has trivially converged to $\hat{s}_{uv}^{(g)} = 0$ for all edges in $L_-$, its expected score will be strictly greater than 0:

$$\mathbb{E}\left[\hat{s}_{uv}^{(g)} \,\middle|\, L_-\right] > 0$$

Therefore, the expected gradient is strictly positive. This positive gradient causes the logit $z_{uv}$ to decrease on average during gradient descent, pushing the score $\hat{s}_{uv}^{(g)}$ towards 0. This completes the proof. $\square$

### B.3 DERIVATION OF BINARIZATION IN THE GRAY ZONE VIA NEIGHBORHOOD INFLUENCE

*Proof of Corollary 1.* The corollary states that the score of an unlabeled edge $(u, v)$ is updated based on the pseudo-labels of its structurally proximate labeled neighbors. We prove this by analyzing the flow of gradients through the shared parameters $\theta$ of the GNN-based guided explainer.

**Step 1: Deriving the Score Update Equation.** The parameters $\theta$ of the guided explainer are updated via gradient descent: $\theta_{t+1} = \theta_t - \eta \nabla_\theta \mathcal{L}_g(\theta_t)$. The first-order Taylor approximation for the change in the score of an unlabeled edge $(u, v) \notin L$ is:

$$\Delta \hat{s}_{uv}^{(g)} = \hat{s}_{uv}^{(g)}(\theta_{t+1}) - \hat{s}_{uv}^{(g)}(\theta_t) \approx \langle \nabla_\theta \hat{s}_{uv}^{(g)}(\theta_t), \theta_{t+1} - \theta_t \rangle = -\eta \langle \nabla_\theta \hat{s}_{uv}^{(g)}, \nabla_\theta \mathcal{L}_g \rangle \tag{9}$$

The loss $\mathcal{L}_g$ is a sum over only the labeled edges in $L$. Therefore, its gradient is:

$$\nabla_\theta \mathcal{L}_g = \sum_{(i,j) \in L} \nabla_\theta l_w(\hat{s}_{ij}^{(g)}, s_{ij}') = \sum_{(i,j) \in L} \frac{\partial l_w}{\partial \hat{s}_{ij}^{(g)}} \nabla_\theta \hat{s}_{ij}^{(g)} \tag{10}$$

Substituting this decomposition back into the expression for $\Delta \hat{s}_{uv}^{(g)}$ yields the equation stated in the corollary:

$$\Delta \hat{s}_{uv}^{(g)} \approx -\eta \sum_{(i,j) \in L} \frac{\partial l_w}{\partial \hat{s}_{ij}^{(g)}} \langle \nabla_\theta \hat{s}_{uv}^{(g)}, \nabla_\theta \hat{s}_{ij}^{(g)} \rangle$$

**Step 2: Analyzing the Components of the Update.** The update is a sum of contributions from all labeled edges. Each contribution is a product of two terms:

1. **The Error Signal** ($\partial l_w / \partial \hat{s}_{ij}^{(g)}$): This scalar term indicates the direction of correction for a labeled edge $(i, j)$. For the weighted binary cross-entropy loss $l_w(\hat{s}, s') = -[w_+ s' \log \hat{s} + (1 - s') \log(1 - \hat{s})]$, its derivative is:

$$\frac{\partial l_w}{\partial \hat{s}_{ij}^{(g)}} = \begin{cases} -w_+/\hat{s}_{ij}^{(g)} < 0 & \text{if } (i,j) \in L^+ \text{ (i.e., } s_{ij}' = 1) \\ 1/(1 - \hat{s}_{ij}^{(g)}) > 0 & \text{if } (i,j) \in L^- \text{ (i.e., } s_{ij}' = 0) \end{cases}$$

   The signal is negative for positive pseudo-labels (pushing the score up) and positive for negative pseudo-labels (pushing the score down).

2. **The Gradient Alignment** ($\langle \nabla_\theta \hat{s}_{uv}^{(g)}, \nabla_\theta \hat{s}_{ij}^{(g)} \rangle$): This inner product measures the influence of a learning signal from $(i, j)$ on the score of $(u, v)$. The score $\hat{s}_{uv}^{(g)}$ is a function of the final node embeddings, which are themselves complex functions of the shared GNN parameters $\theta$ and the graph topology. The gradient $\nabla_\theta \hat{s}_{uv}^{(g)}$ reflects the sensitivity of the score to changes in these shared parameters.

   This alignment term is large and positive if and only if the computational graphs for $(u, v)$ and $(i, j)$ have significant overlap. In a GNN, this occurs when edges are structurally proximate (e.g., they are incident to the same node or are 1-hop neighbors). The shared weights $\theta$ in the message-passing layers are updated based on local neighborhood structures, so nearby edges will have highly correlated gradients with respect to $\theta$. Conversely, for distant edges, the alignment is near zero.

**Step 3: Synthesis of Neighborhood Influence.** We analyze the net effect on $\Delta \hat{s}_{uv}^{(g)}$ based on the dominant pseudo-label in the neighborhood of $(u, v)$.

- **Case 1: Positively Dominated Neighborhood.** Assume the labeled edges structurally proximate to $(u, v)$ are mostly in $L^+$. The sum for the update will be dominated by these positive neighbors. For each such neighbor $(i, j) \in L^+$, its contribution to the sum is a product of a negative error signal and a large positive alignment term. This product is negative. The total sum is therefore a large negative value. The final change is $\Delta \hat{s}_{uv}^{(g)} \approx (-\eta) \times$ (large negative sum) $> 0$. Thus, the score $\hat{s}_{uv}^{(g)}$ increases.

- **Case 2: Negatively Dominated Neighborhood.** Assume the labeled neighbors of $(u, v)$ are mostly in $L^-$. For each such neighbor $(i, j) \in L^-$, its contribution is a product of a positive error signal and a large positive alignment term. This product is positive. The total sum is a large positive value. The final change is $\Delta \hat{s}_{uv}^{(g)} \approx (-\eta) \times$ (large positive sum) $< 0$. Thus, the score $\hat{s}_{uv}^{(g)}$ decreases.

In conclusion, the GNN architecture inherently propagates supervision from labeled edges to their unlabeled neighbors through updates to its shared weights. An unlabeled edge's score is pushed towards the consensus pseudo-label of its local neighborhood, providing a formal mechanism for the binarization of scores in the gray zone. □

### B.4 ASSUMPTIONS FOR SECTION 3.3

**A1 (Calibration/monotonicity).** Let $\eta_{uv} := \Pr(s_{uv} = 1 \mid G)$. There exists $\delta \in [0, \frac{1}{2})$ such that $\sup_{(u,v) \in \mathcal{E}} |\hat{s}_{uv}^{(b)} - \eta_{uv}| \le \delta$.

**A2 (Low-density decision boundary).** There exist $C > 0, \kappa > 0$ such that $\Pr\left(|\eta_{uv} - \frac{1}{2}| \le \gamma\right) \le C \gamma^\kappa$ for all $\gamma > 0$.

### B.5 RISK AND EMPIRICAL OBJECTIVE

Denote by $L \subseteq \mathcal{E}$ the labeled (confidence zone) edges obtained from the skew-aware quantile rule in Section 3.1, and by $\mathcal{N}$ an edge-neighborhood system (e.g., line-graph adjacency) used only for analysis.

Let $\mathcal{D}$ be a distribution over graphs $G = (\mathbf{X}, \mathbf{A})$ with edge set $\mathcal{E}$. For each edge $(u, v) \in \mathcal{E}$, the (unknown) target is the binary label $s_{uv} \in \{0, 1\}$. Let $\ell_w$ be the weighted binary cross entropy. We consider the population *tail risk*:

$$R_{\text{tail}}(h^{(g)}) = \mathbb{E}_{G \sim \mathcal{D}}\Big[\frac{1}{|L|} \sum_{(u,v) \in L} \ell\big(\hat{s}_{uv}^{(g)}, s\big)\Big],$$

which matches our evaluation protocol on confident edges. $h^{(g)}$ is trained with the *supervised-on-tails* objective:

$$\widehat{R}_{\text{sup}}(h^{(g)}) = \frac{1}{|L|} \sum_{e \in L} \ell_w\big(\hat{s}_{uv}^{(g)}, s'\big), \qquad \ell_w(\hat{s}, s') = -\big[w_+ \, s' \log \hat{s} + (1 - s') \log(1 - \hat{s})\big]. \quad (11)$$

**Tail purity (noise upper bounds).** Define the tail noise rates $\varepsilon_+ := \Pr(s = 0 \mid \hat{s}_{uv}^{(b)} \ge t_U)$ and $\varepsilon_- := \Pr(s = 1 \mid \hat{s}_{uv}^{(b)} \le t_L)$, where edge $(u, v)$ is arbitrarily chosen. Under a mild calibration/monotonicity condition on $\hat{s}_{uv}^{(b)}$ (Appendix B.4), one has $\varepsilon_+ \le 1 - t_U + \delta$ and $\varepsilon_- \le t_L + \delta$, so both tails are majority-correct when $t_U > 1/2 + \delta$ and $t_L < 1/2 - \delta$.

**Generalization bound on confident edges.** Let $\mathcal{H}_g$ denote the hypothesis class of guided explainers and $\mathfrak{R}_{n_\ell}(\mathcal{H}_g)$ be the empirical Rademacher complexity Bartlett and Mendelson (2002); Mohri et al. (2018) of $\mathcal{H}_g$ over $n_\ell$ labeled edges, where $n_\ell := |L|$.

**Theorem 1** (Generalization of 3SG-Explainer on the confidence zone). *With probability at least $1 - \delta$ over the draw of training graphs and the confidence zone selection, every $h^{(g)} \in \mathcal{H}_g$ satisfies*

$$R_{\text{tail}}(h^{(g)}) \le \underbrace{\widehat{R}_{sup}(h^{(g)})}_{\text{empirical supervised risk}} + \underbrace{2\,\mathfrak{R}_{n_\ell}(\mathcal{H}_g) + c_1 \sqrt{\frac{\log(1/\delta)}{n_\ell}}}_{\text{statistical complexity}} + \underbrace{c_2\big(\varepsilon_+ + \varepsilon_-\big)}_{\text{pseudo-label noise}}. \quad (12)$$

*Here $c_1, c_2 > 0$ are universal constants.*

*Sketch.* Condition on $L$ and apply uniform convergence for bounded Lipschitz losses to control the first two terms; bound the difference between true labels and pseudo-labels on $L$ by the tail noise rates. A complete proof is deferred to Appendix B.6.

Compared to a base MLP explainer, the GNN structure (message passing, parameter sharing, permutation invariance) reduces $\mathfrak{R}_{n_\ell}(\mathcal{H}_g)$ at a matched hidden dimension, yielding a tighter statistical term. In practice this translates into better generalization on the confident edges.

### B.6 PROOF OF THEOREM 1

We use the clipped BCE $\ell_\tau(\hat{s}, y)$ with $\hat{s} \in [\tau, 1 - \tau]$; it is $B_\tau$-bounded and $L_\tau$-Lipschitz.

**Step 1: Uniform convergence on the selected tails.** Condition on the data-dependent tail set $L$ (measurable w.r.t. base scores). By symmetrization and contraction, for any $\delta \in (0, 1)$, with probability at least $1 - \delta/2$,

$$\mathbb{E}\Big[\frac{1}{n_\ell} \sum_{e \in L} \ell\big(\hat{s}_{uv}^{(g)}, s_{uv}'\big)\Big] \leq \frac{1}{n_\ell} \sum_{e \in L} \ell\big(\hat{s}_{uv}^{(g)}, s_{uv}'\big) + 2\,\mathfrak{R}_{n_\ell}(\mathcal{H}_g) + c_1 \sqrt{\frac{\log(2/\delta)}{n_\ell}}.$$

**Step 2: Noise correction on the tails.** On $L_+ = \{(u, v) : \hat{s}_{uv}^{(b)} \geq t_U\}$ put $s' = 1$ and on $L_- = \{(u, v) : \hat{s}_{uv}^{(b)} \leq t_L\}$ put $y' = 0$. Since $\ell_\tau$ is bounded by $B_\tau$,

$$\big|\ell_\tau(\hat{s}_{uv}^{(g)}, s) - \ell_\tau(\hat{s}_{uv}^{(g)}, s')\big| \leq B_\tau\,\mathbf{1}\{s \neq s'\}.$$

Taking expectations and using $\Pr(s_{uv} \neq s_{uv}' \mid L_+) = \varepsilon_+$ and $\Pr(s_{uv} \neq s_{uv}' \mid L_-) = \varepsilon_-$ gives

$$\mathbb{E}\Big[\frac{1}{n_\ell} \sum_{(u,v) \in L} \ell(\hat{s}_{uv}^{(g)}, s_{uv})\Big] \leq \mathbb{E}\Big[\frac{1}{n_\ell} \sum_{e \in L} \ell(\hat{s}_{uv}^{(g)}, s_{uv}')\Big] + c_2(\varepsilon_+ + \varepsilon_-).$$

**Step 3: Combine.** Combine Steps 1–2 and remove the clipping constants into $c_1, c_2$ to obtain equation 12.

### B.7 EXTENSION TO FULL-EDGE RISK

If one wishes to upper bound the full-edge risk $R(h^{(g)}) = \mathbb{E}\big[\frac{1}{|\mathcal{E}|} \sum_{(u,v) \in \mathcal{E}} \ell(\hat{s}_{uv}^{(g)}, s)\big]$, a simple coverage term $B_\tau \frac{|\mathcal{E} \setminus L|}{|\mathcal{E}|}$ can be added (vacuous but assumption-free); alternatively, introducing an unlabeled consistency regularizer (e.g., a Laplacian penalty) during analysis yields

$$R(h^{(g)}) \leq \frac{|L|}{|\mathcal{E}|}\,R_{\text{tail}}(h^{(g)}) + \lambda\,\Omega_{\text{Lap}}(h^{(g)}) + c_3\,\frac{\Lambda}{|\mathcal{N}|},$$

under a smoothness prior on the target function.

**Remark 1** (No unlabeled regularizer is used in training). We do not use any unlabeled (e.g., Laplacian) term in training. *Any $\Omega_{Lap}$ that appears in the Appendix is an analysis device for bounding the contribution of unlabeled edges when one extends the bound from $R_{\text{tail}}$ to the full-edge risk. In all experiments we set $\lambda = 0$ and optimize only equation 11.*

Note that our algorithm sets $\lambda = 0$ (Remark 1), and this extension is theoretically sound.

### B.8 VC-BASED COMPARISON BETWEEN BASE AND GUIDED EXPLAINERS

We briefly recall the definition of VC dimension (Bartlett et al. (2019); Shalev-Shwartz and Ben-David (2014)) , which provides a principled way to quantify the capacity of a function class and analyze its generalization behavior.

**Definition 1** (VC Dimension). *The Vapnik–Chervonenkis (VC) dimension of a function class $\mathcal{H}$ is defined as the size of the largest set of points that can be shattered by $\mathcal{H}$. A set is said to be shattered if, for every possible binary labeling of the set, there exists a function in $\mathcal{H}$ that realizes it.*

**Definition 2** (Generalization gap). *Let $\mathcal{X}$ and $\mathcal{Y}$ be feature and label spaces, respectively. For a function $h : \mathcal{X} \to \mathcal{Y}$, a bounded loss $\mathcal{L}$,[1] and a dataset $\{(x_i, y_i)\}_{i=1}^m$, the generalization gap is*

$$\Delta(h) := \mathbb{E}_{(x,y)\sim\mathcal{P}}[\mathcal{L}(h(x), y)] - \frac{1}{m}\sum_{i=1}^{m}\mathcal{L}(h(x_i), y_i).$$

**Lemma 3** (VC-dimension generalization bound). *Let hypothesis class $\mathcal{H}$ have VC dimension $d_{\mathrm{VC}}$ and $\mathcal{L}$ be bounded. Then, for any $\delta \in (0, 1)$, with probability at least $1 - \delta$, every $h \in \mathcal{H}$ satisfies*

$$\Delta(h) \leq C\sqrt{\frac{d_{\mathrm{VC}} + \log(1/\delta)}{m}},$$

*for a universal constant $C > 0$.*

**Lemma 4** (VC-dimension of piecewise-linear MLPs). *Consider an MLP with piecewise-linear activations, $W$ trainable parameters and $L$ layers. Its VC dimension satisfies $d_{\mathrm{VC}} = \mathcal{O}(W L \log W)$.*

**Lemma 5** (Structure-aware VC control for GNNs). *For a GNN with ReLU nonlinearities and message-passing with a fixed aggregation operator, the VC dimension is upper-bounded by a function of the final hidden width $d_{L-1}$ and structural factors. In particular, for matched widths, a GNN typically admits a strictly smaller $d_{\mathrm{VC}}$ than a plain MLP due to these structural priors.*

**Theorem 2** (VC-dimension based generalization gaps). *Let $h^{(b)}$ be the base (MLP) explainer and $h^{(g)}$ the guided (GNN) explainer. Let $m$ be the number of training edges used for $h^{(b)}$, and $n_\ell$ the number of pseudo-labeled (confidence-zone) edges used for $h^{(g)}$. Then, for any $\delta \in (0, 1)$, with probability at least $1 - \delta$,*

$$\Delta(h^{(b)}) \leq \tilde{C}\sqrt{\frac{W L \log W + \log(1/\delta)}{m}}, \tag{13}$$

$$\Delta(h^{(g)}) \leq \tilde{C}\sqrt{\frac{d_{L-1} + \log(1/\delta)}{n_\ell}}, \tag{14}$$

*for a universal constant $\tilde{C} > 0$.*

**Discussion (why the guided gap is tighter).** By equation 13–equation 14, the base MLP gap scales with $W L \log W$, while the guided GNN gap is controlled essentially by the last-layer width $d_{L-1}$. Because message passing, parameter sharing, and permutation invariance reduce the effective capacity, the guided bound is typically *tighter*, even though it uses only $n_\ell \leq m$ confident samples. This matches the empirical reliability improvements we observe for 3SG-Explainer.

## C  DATASETS

- **BA-3motifs**: The BA-3motifs dataset consists of 3,000 graphs, each constructed by attaching one of three motif types—house, cycle, or grid—to a Barabási-Albert (BA) base graph. The class label is determined by the type of attached motif. Ground-truth edge-level explanations are defined as the set of edges comprising the inserted motif, which highlight the key structural patterns responsible for the graph label.

- **Mutagenicity**: The Mutagenicity dataset is categorized into two classes based on their mutagenic properties. The ground-truth explanations are $NH_2$ and $NO_2$, which have causal relations to the mutagenicity. Unlike conventional experiments that use the entire dataset, we selectively include only the subset of molecules for which edge-level ground-truth annotations are available (a subset of 4,337 graphs). We adopt this restricted setting because, when feasible, using verified ground-truth explanations provides a more realistic and challenging benchmark for evaluating model interpretability. In particular, our experiments are conducted only on the subset of graphs with available edge-level labels.

- **Fluoride-Carbonyl**: The Fluoride-Carbonyl dataset is labeled as positive if a molecule contains a fluoride ($F^-$) and a carbonyl (C=O) functional group. The ground-truth explanations consist of combinations of fluoride atoms and carbonyl groups. Similar to Mutagenicity, we perform experiments exclusively on the subset of graphs for which edge-level ground-truth annotations are provided.

---

[1]In our setting, the clipped BCE $\ell_\tau(\hat{s}, y)$ with $\hat{s} \in [\tau, 1 - \tau]$ is bounded and hence satisfies this assumption.

- **MNIST-75sp**: We use the MNIST-75sp superpixel dataset, which transforms 80,000 digit images into graphs where nodes represent superpixels and edges denote spatial adjacency. Each graph is labeled according to one of ten digit classes. Following prior works, ground-truth important edges correspond to strokes that contribute to digit recognition, identified via heuristics or annotation. For computational efficiency, we randomly sample 10% of the full dataset in our experiments.

- **Benzene**: The Benzene dataset contains 12,000 molecular graphs extracted from the ZINC15 database and is labeled according to the presence of benzene rings. The task is to determine whether an aromatic benzene ring exists within each molecule, and edge-level ground-truth explanations are defined as the set of bonds constituting each benzene ring. We restrict our experiments to the subset of molecules with available edge-level annotations to ensure interpretable evaluation setting.

Table 6: Performance of the GNN models and statistics across different datasets.

| Metric | BA-3motifs | Mutagenicity | Fluoride-Carbonyl | MNIST-75sp | Benzene |
|---|---|---|---|---|---|
| **GNN Type** | GIN | GCN | GCN | GCN | GIN |
| **Val Accuracy** | 0.9750 | 0.8651 | 0.9069 | 0.8333 | 0.9400 |
| **Test Accuracy** | 0.9725 | 0.8462 | 0.9002 | 0.8333 | 0.9392 |
| **Total Graphs** | 3000 | 1356 | 8671 | 7988 | 12000 |
| **Avg. Nodes** | 21.92 | 27.57 | 21.36 | 70.55 | 20.57 |
| **Avg. Edges** | 14.52 | 28.37 | 22.68 | 295.13 | 21.81 |
| **Avg. Degree** | 1.31 | 2.05 | 2.12 | 8.29 | 2.11 |
| **Density** | 0.07 | 0.09 | 0.11 | 0.12 | 0.11 |
| **Edge-GT Rate** | 0.65 | 0.10 | 0.04 | 0.26 | 0.19 |

## D  BASELINES

- **PGExplainer**: PGExplainer generates edge-level explanations by learning a parameterized function that estimates the importance of each edge in the input graph. It employs a neural network to predict edge probabilities, from which subgraphs are sampled via a continuous relaxation. The explainer is trained to preserve the original GNN's predictions on the sampled subgraphs.

- **ReFine**: ReFine adopts a pre-training and fine-tuning paradigm. It first learns class-wise attribution models to produce global saliency maps, and then fine-tunes these maps to generate instance-specific explanatory subgraphs. This two-stage approach aims to capture both global structure and local fidelity.

- **TAGE**: TAGE is a task-agnostic explanation framework that decouples the explanation process into two phases. It first trains a self-supervised embedding explainer to identify subgraphs preserving mutual information with original graph embeddings. At inference, a gradient-based explainer introduces task-specific conditions to guide subgraph selection.

- **MixupExplainer**: MixupExplainer tackles the distribution shift that arises when explanation subgraphs differ significantly from the graphs used to train the GNN. It addresses this by mixing each explanatory subgraph with a label-independent subgraph from another graph, resulting in synthetic graphs that better match the original training distribution. This improves the fidelity of explanations.

- **ProxyExplainer**: ProxyExplainer addresses the out-of-distribution issue in GNN explanations by generating in-distribution proxy graphs using a variational autoencoder. This generative strategy ensures that explanations remain faithful and reliable by aligning the proxy graphs with the original training distribution.

- **EiG-Search**: EiG-Search aims to address the limitations of fixed-size and node-induced explanations by shifting to an edge-induced search framework. It computes importance scores for all edges using a linear-gradient approximation and then performs an efficient search to identify a subgraph that best explains the prediction. As a training-free and non-GIB method, it produces adaptive and instance-specific explanations without learning an auxiliary model.

- **GOAt**: GOAt provides architecture-agnostic explanations by analytically decomposing the GNN forward pass into interpretable product terms and attributing contributions directly to edges. This yields deterministic edge scores that capture their effect on the final output. As a non-GIB and

training-free approach, GOAt generates stable explanations without relying on mask optimization or additional training.

# E  EXPERIMENTAL DETAILS

**Metrics for distributional analysis.**  To quantitatively assess the shape of the edge score distribution, we employ two metrics.

We first measure the degree of bimodality, defined as

$$\text{Bimodality}(p_1, \ldots, p_n) = \frac{\frac{\left(\mu_3/\mu_2^{3/2}\right)^2 + 1}{\mu_4/\mu_2^2} - \frac{1}{3}}{1 - \frac{1}{3}}, \quad \mu_k = \frac{1}{n}\sum_{i=1}^{n}(p_i - \bar{p})^k.$$

This normalized form of the bimodality coefficient yields values closer to 1 when the distribution exhibits clearer separation of modes. Complementing this, we also introduce the binarization score, which directly quantifies how strongly the scores polarize toward the two extremes.

$$\text{Binarization score}(p_1, \ldots, p_n) = \frac{1}{n}\sum_{i=1}^{n}(2p_i - 1)^2.$$

This measures the concentration of scores toward the binary extremes $\{0, 1\}$. Values close to 1 reflect strongly polarized distributions, while values near 0 indicate scores clustered around 0.5.

# F  EXPERIMENTAL DETAILS

**Hyperparameter search**  We conducted all experiments on a server equipped with eight NVIDIA RTX A6000 GPUs, each with 48GB of memory. We use the Adam optimizer for training and adopt a dataset split ratio of either 7:1.5:1.5 for training, validation, and testing, depending on the dataset.

- Batch size: 16
- Number of GNN layers: 4
- Hidden dimension: 64
- Learning rate: $\{10^{-3}, 5 \times 10^{-4}, 10^{-4}\}$

We evaluate each hyperparameter setting across 10 random seeds and report the average performance along with standard deviations. We use a GCN as architecture for the guided explainer on BA-3motifs, Mutagenicity, and MNIST-75sp, while a GIN is employed for Fluoride-Carbonyl. We performed grid search over the following hyperparameter ranges:

- $\alpha_0$: $\{0.05, 0.1, 0.2\}$
- $c$: $\{0.05, 0.1\}$
- $w^+$: $\{1.0, 3.0, 5.0\}$

# G  ROBUSTNESS TO HYPERPARAMETER CHOICES

We additionally examined the stability and sensitivity of our model to the three primary hyperparameters introduced in Appendix F, $\alpha_0$, $c$, and $w^+$. For each dataset, we explored the full hyperparameter grid, consisting of $3 \times 2 \times 3$ combinations for each of the two rounds, resulting in a total of $18 \times 18 = 324$ configurations.

Figure 5 visualizes the performance variation across all hyperparameter combinations for each dataset. The blue square indicates the configuration reported in Table 1, the black horizontal line denotes the median performance over the entire grid, and the orange marker represents the performance of the base explainer. Importantly, our evaluation considers not only the best-performing points but also the overall distribution; therefore, we do not always report the maximum value but instead emphasize stable and robust configurations.

Table 7: Training and inference time cost (in seconds).

| Method | Training time (s) | | | | | Inference time (s) | | | | |
|---|---|---|---|---|---|---|---|---|---|---|
| | BA3 | MUTAG | MNIST | FC | Benzene | BA3 | MUTAG | MNIST | FC | Benzene |
| PGExplainer | 14.32 | 9.63 | 48.62 | 43.95 | 29.73 | 0.19 | 0.04 | 0.66 | 0.29 | 0.08 |
| ReFine | 419.51 | 511.68 | 1124.89 | 442.55 | 217.43 | 0.49 | 0.47 | 1.20 | 1.26 | 0.57 |
| TAGE | 5.92 | 7.51 | 85.49 | 30.92 | 8.45 | 0.20 | 0.04 | 0.55 | 0.28 | 0.08 |
| MixupExplainer | 1327.37 | 526.07 | 8075.48 | 2962.07 | 290.38 | 0.18 | 0.10 | 0.87 | 0.52 | 0.21 |
| ProxyExplainer | 137.50 | 32.81 | 113.17 | 391.93 | 87.81 | 0.17 | 0.04 | 0.82 | 0.30 | 0.08 |
| **3SG-Explainer** | **5.41** | **6.69** | **71.17** | **5.08** | **7.87** | **0.04** | **0.03** | **0.54** | **0.03** | **0.07** |

Across all datasets, the aggregated results show that the AUC of 3SG-Explainer exceeds that of the base explainer for the vast majority of hyperparameter combinations. This demonstrates that the self-guidance mechanism is intrinsically robust and does not rely heavily on fine-tuned hyperparameters to achieve significant improvements.

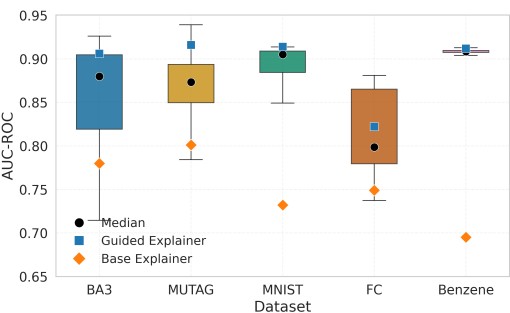

Figure 5: Hyperparameter analysis summarized with boxplots across datasets. Each boxplot illustrates how the performance varies across combinations of $\alpha_0$, $c$, and $w$. This consolidated visualization provides a clearer overview of robustness across a wide range of hyperparameter configurations.

## H  LOW COMPUTATIONAL COST

To assess the computational cost of our approach, we evaluate the training and inference efficiency of the proposed module. Training time is measured end-to-end and includes validation performed every epoch. For models whose training extends beyond 30 epochs, we apply early stopping with a patience of 30 to ensure flexibility across datasets. For 3SG-Explainer, the reported training time includes both rounds (Round 1 and Round 2) as well as the intermediate pseudo-labeling steps. Inference time is measured on the testset by computing the total time required to evaluate all graphs within the split.

As shown in Table 7, our analysis focuses on the additional training time introduced by the multi-round refinement. The guided explainer is trained on top of a pre-trained base explainer (PGExplainer for our experiments), and the extra cost added by additional rounds remains relatively small enough across datasets. In other words, the multi-round self-guidance does not impose a meaningful overhead beyond the cost of training the base explainer itself. In contrast, inference cost can be compared directly across methods. Across all datasets, our method achieves efficient inference speeds better than the existing baselines, indicating that our method introduces no noticeable overhead during evaluation.

Importantly, even on MNIST-75sp, our largest and densest dataset, 3SG-Explainer maintains cost-effectiveness. This indicates that our method scales well with increasing graph size and density, suggesting that scalability is not a limiting factor in practical deployment. Overall, despite the presence of multiple rounds, our method achieves efficient training and fast inference, making it a practical and scalable framework for real-world applications.

# I  STABILITY UNDER IMPERFECT PSEUDO-LABELS

To evaluate the stability of our approach under noisy confidence score, we conduct a controlled noise-injection study. In this setting, a fixed percentage of pseudo-labels is randomly flipped to the opposite label at each round, while all hyperparameters remain identical to the main setup.

As shown in Figure 6, the AUC remains largely unchanged even as the noise ratio increases, indicating that predictive performance is robust to perturbations in the pseudo-labels. The bimodality and binarization scores change only slightly across noise levels and keeps high degree of polarization. Interestingly, in some cases a small amount of noise nudges the mask distribution toward a more polarized and stable form, which can be interpreted as the injected perturbations regularizing the inherent noise already present in the pseudo-labels. Overall, these observations indicate that 3SG-Explainer does not rely heavily on perfectly clean pseudo-labels; even with nontrivial noise, it consistently produces accurate explanations with well-polarized edge masks.

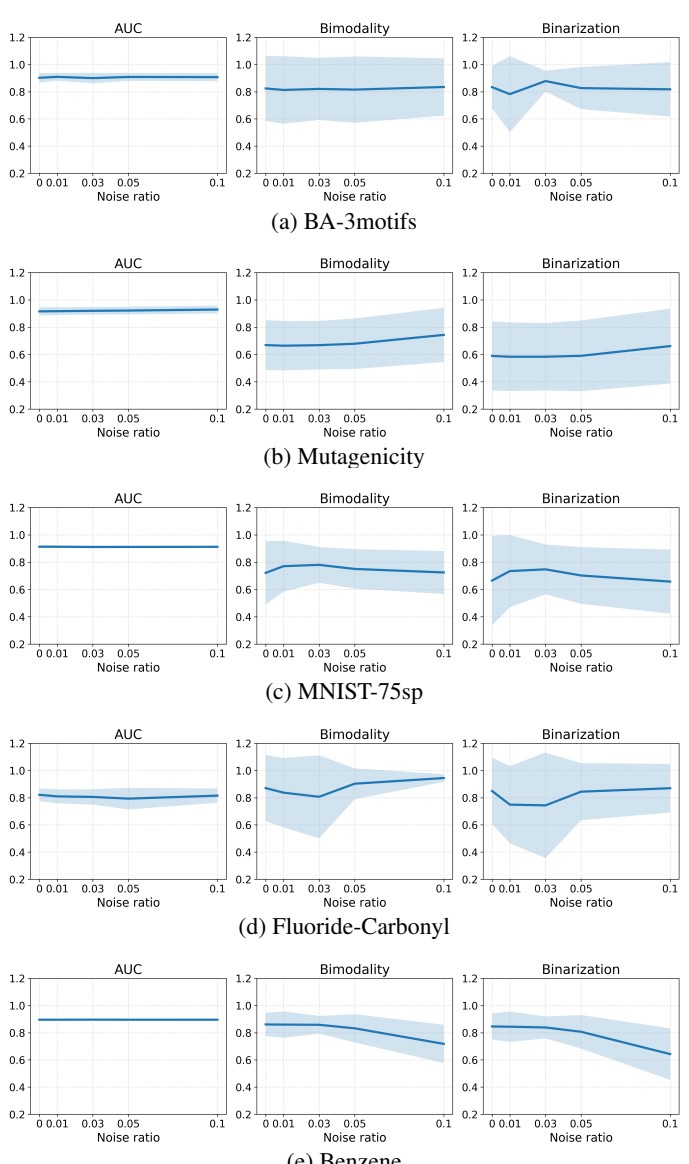

Figure 6: Effect of injected pseudo-label noise on explanation quality across the datasets.

## J  ADDITIONAL QUALITATIVE RESULTS

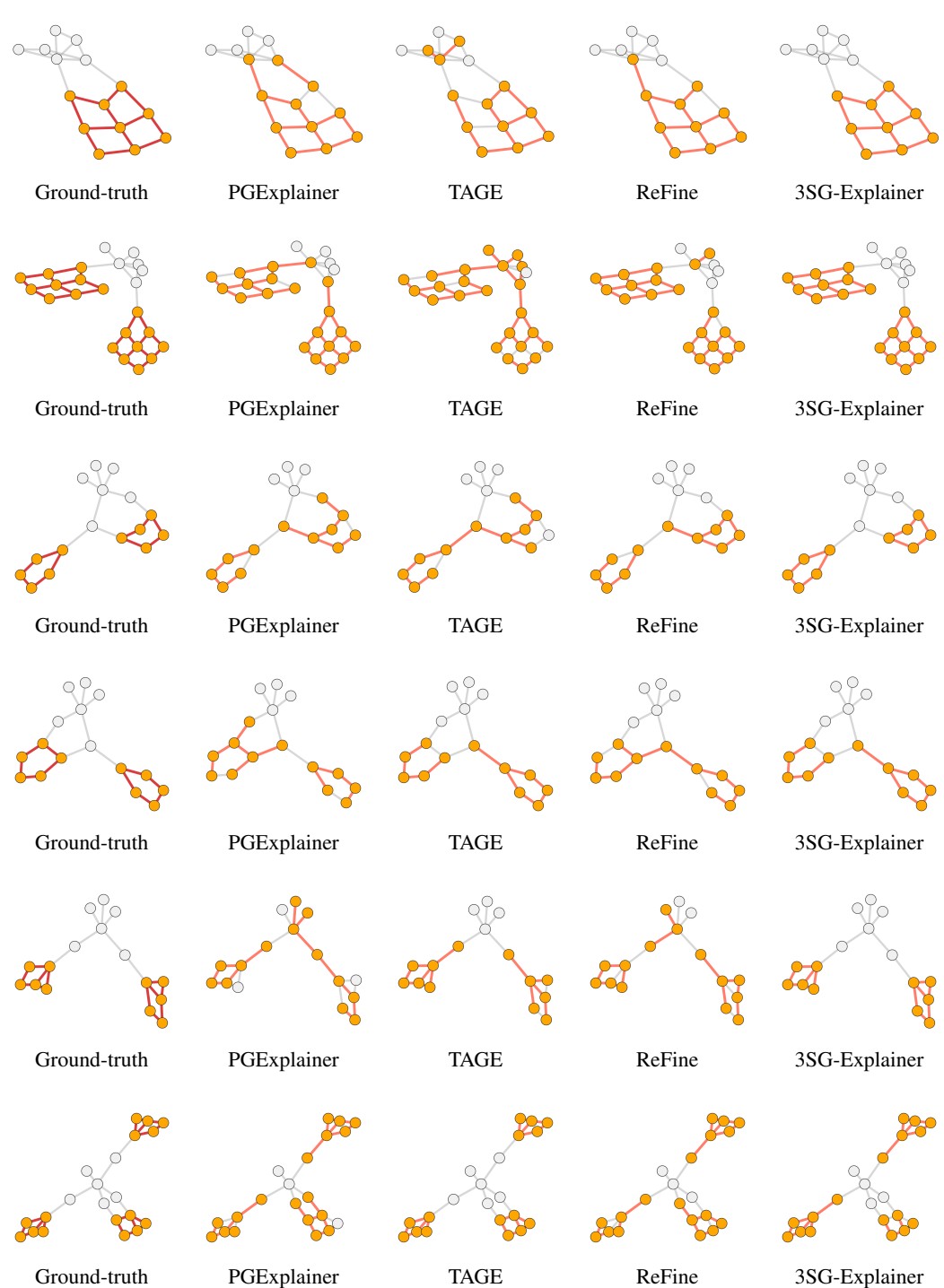

Figure 7: Qualitative comparison on BA-3motifs. Each row shows one example graph. The BA-3motifs dataset contains three distinct motif types, and our examples cover all of them. In these graphs, the proportion of important substructures is relatively high compared to the total number of edges. While baselines often fail to recover all motif edges exactly, 3SG-Explainer successfully corrects these errors through self-guidance and produces a more complete and faithful explanation.

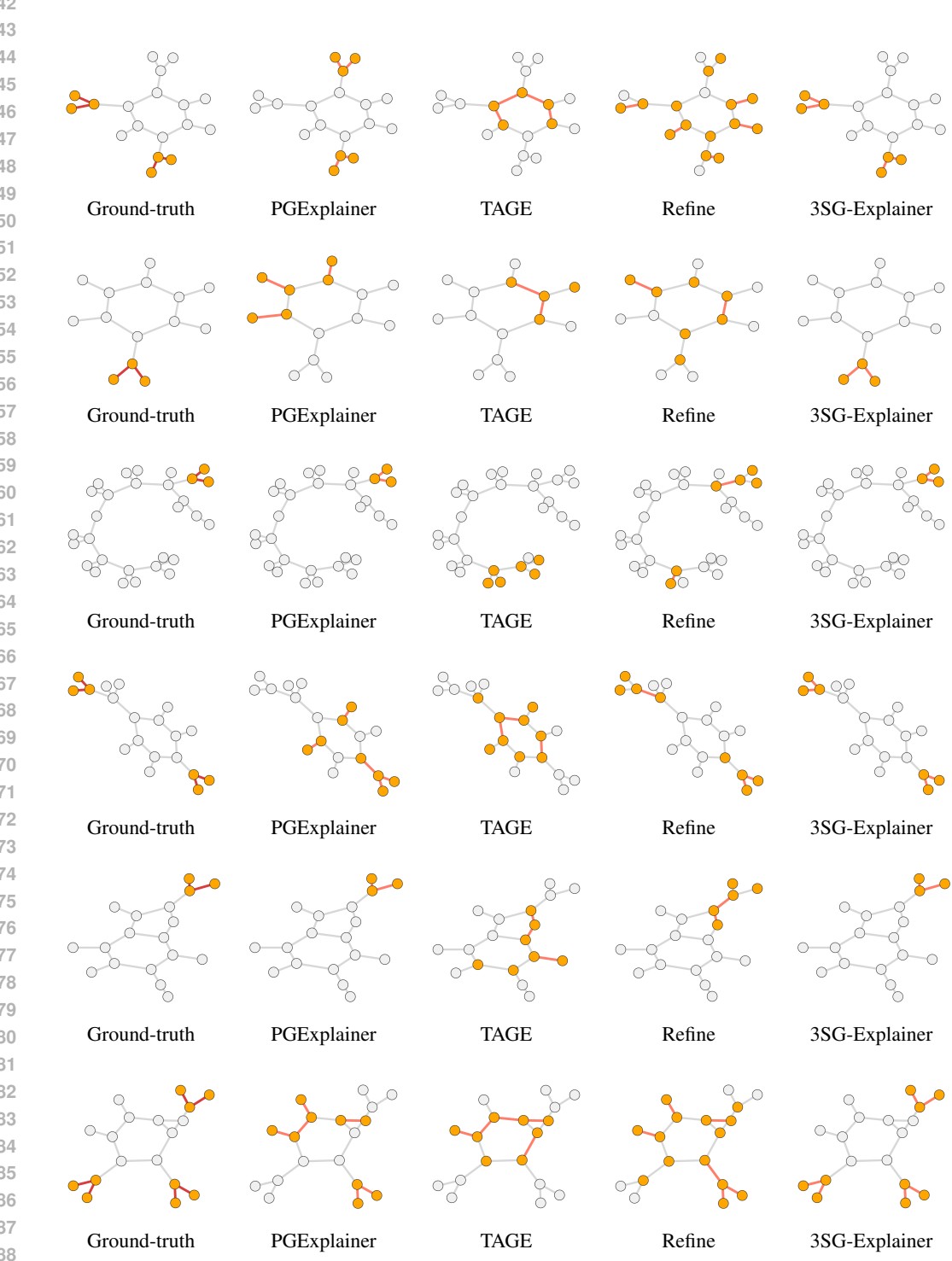

Figure 8: Qualitative comparison on Mutagenicity. Each row shows one molecule. Mutagenicity is a real-world molecular dataset whose graphs exhibit diverse backbone structures with small functional subgraphs attached. Although baselines often capture only part of the functional group, 3SG-Explainer more accurately isolates the relevant subgraph by leveraging self-guidance, leading to a clearer and more complete explanation.

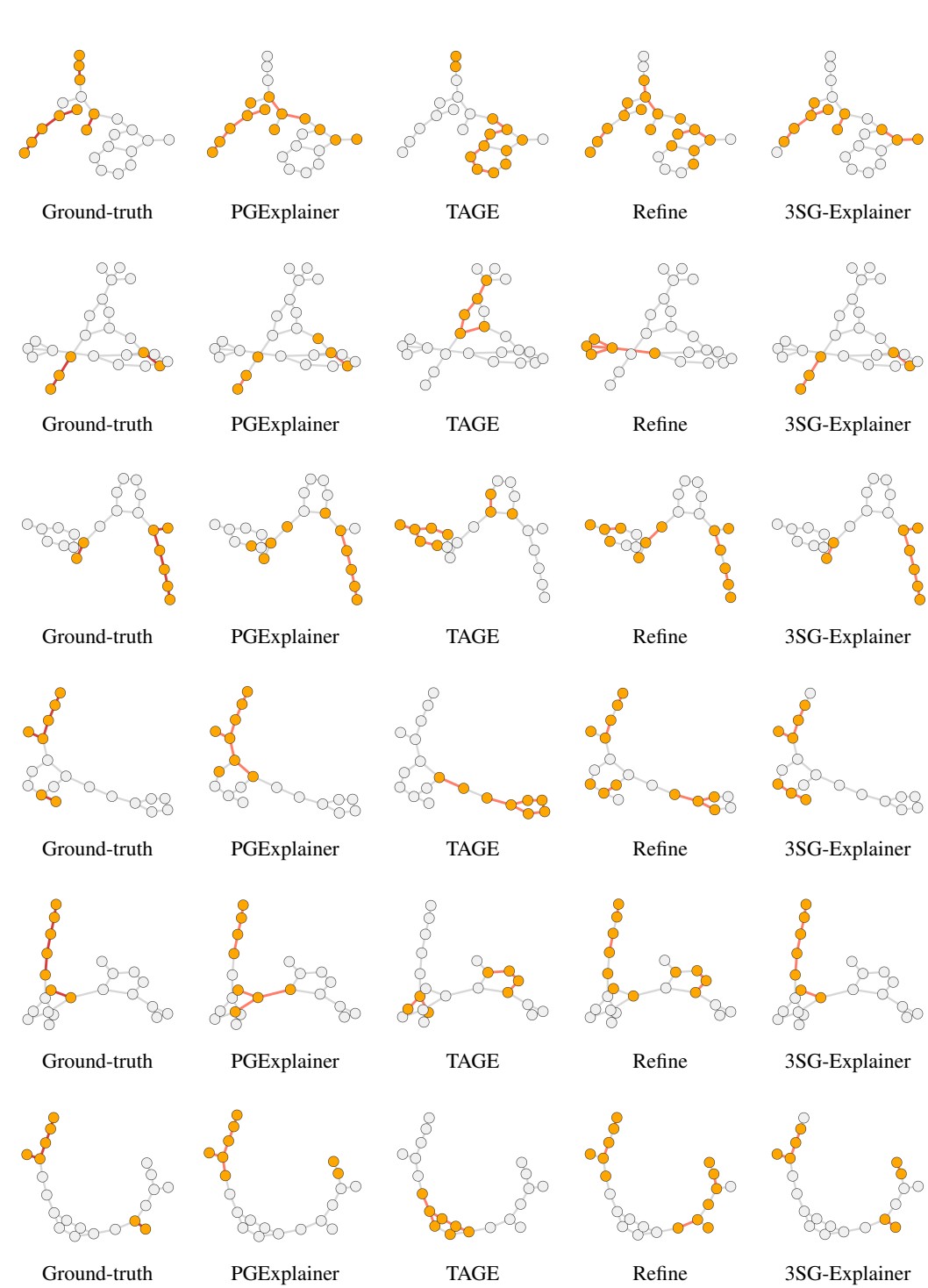

Figure 9: Qualitative comparison on Fluoride–Carbonyl. Each row shows one molecule. The Fluoride–Carbonyl dataset contains well-defined chemical interaction patterns, but accurately isolating the functional subgraph remains challenging due to subtle structural cues. While 3SG-Explainer still struggles to perfectly refine the important substructure, it consistently provides more ground-truth–aligned patterns compared to the baselines, demonstrating a clear improvement in explanatory quality.

