# OpenReview forum: "Self-Guided Explanation for Graph Neural Networks with Semi-Supervision"
_ICLR.cc/2026/Conference — Submitted to ICLR 2026_

### Official Review · Reviewer_KmWJ · 2025-10-30

**Soundness:** 2
**Presentation:** 2
**Contribution:** 3
**Rating:** 4
**Confidence:** 4

**Summary:**

This paper essentially introduces a post-hoc binarization framework for GNN explainers. It extracts high-confidence pseudo-labels from the continuous edge scores of existing GIB-based explainers (e.g., PGExplainer) and trains a GCN to propagate these binary signals across the graph structure. Through iterative self-guided refinement, the edge importance scores become increasingly polarized toward 0 or 1. The theoretical analysis shows that conventional GIB optimization objectives tend to converge to intermediate, non-binary values, while their semi-supervised BCE training drives the scores away from 0.5 and spreads the binary trend to unlabeled edges. As a result, the explanations become more distinct and human-interpretable.

**Strengths:**

1. The motivation for self-guided explanation is well-grounded. The paper identifies a real limitation in current GNN explainers, where continuous, non-binarized edge importance scores hinder interpretability, and they aim to address it in a principled way.
2. The proposed semi-supervised framework that propagates high-confidence binary signals through message passing is conceptually simple yet effective. It aligns well with the structure of GNNs and provides a natural mechanism for refining explanations.
3. Theoretical analysis helps explain why traditional GIB-based objectives tend to produce ambiguous scores, and why the proposed semi-supervised BCE loss yields more polarized distributions. This strengthens the conceptual soundness of the approach.

**Weaknesses:**

### 1. Limited scope of base explainers.
The paper only focuses on optimization-based explainers within the GIB family. However, search-based methods such as SubgraphX and attribution-based methods such as GOAt or GNN-LRP also output confidence scores. It should be possible to use these methods to obtain the initial pseudo-binary edge labels for the first round. Moreover, since GIB-based explainers inherently produce confidence scores clustered around the middle range, while other types of explainers may already yield more polarized edge scores, the need for additional binarization might not arise. Therefore, comparing this method against non-GIB explainers becomes even more necessary. The paper does not discuss this issue or include such comparisons, which makes the evaluation less convincing.
### 2. Pseudo-label noise accumulation.
The authors only prove that their GCN-based guided explainer can converge to binary predictions when the pseudo-label accuracy is greater than 0.5. However, they do not verify whether these binarized labels are truly reliable. There is a risk of pseudo-label drift: if the base explainer’s initial scores are inaccurate, the resulting pseudo-labels may be wrong and such noise could accumulate and be reinforced through self-guided iterations. Although the authors introduce a confidence interval to exclude uncertain edges, this does not guarantee the correctness of high-confidence edges.
### 3. Threshold sensitivity.
Threshold sensitivity. The thresholds are determined by the skewness-based quantile rule, but they still depend on manually set parameters $\alpha$ and c. Since the score distributions vary significantly across datasets, the chosen thresholds may not generalize well. Without sensitivity or robustness analysis, the stability of the pseudo-labeling process remains uncertain.
### 4. Potential propagation bias.
Propagation bias. The guided explainer propagates confidence signals through message passing. If the high-confidence regions themselves are biased or incorrect (for example, due to structural imbalance in molecular graphs), this bias could be amplified and spread across the graph. The paper does not analyze this potential issue.
### 5. Interpretability paradox.
There is a conceptual concern: the method introduces another GCN to explain a GCN-based model. This design may appear recursive, where they use a neural network to interpret another neural network, and somewhat undermines the goal of interpretability.
### 6. Experimental limitations.
The paper does not evaluate fidelity, lacks qualitative visualization of the generated explanations, and uses a limited number of datasets. The real-world interpretability, particularly on molecular datasets, remains unclear.

**Questions:**

see weaknesses.

---

> ### Author Response · Authors · 2025-11-21
>
> We thank the reviewer for the valuable comments. We address Weaknesses 2 and 4 together to clarify the robustness of our method. We hope the following response adequately addresses the concerns raised.
>
> **[W1] Additional experiments with non-GIB baselines**
>
> Our method is not restricted to GIB-based explainers. Because it operates as a post-hoc refinement module that only requires confidence scores, it can be applied to any explainer as long as the method outputs edge-level importance scores. This design enables the framework to naturally accommodate a wide range of explanation techniques beyond the GIB family, which is a key advantage of our module.
>
> To broaden the methodological coverage in the revised paper, we additionally include two non-GIB explainers, Eig-Search [1] and GOAt [2]. Incorporating these methods expands the evaluation beyond the previously considered GIB-based approaches. The detailed results for these new baselines can be found in **Tables 1 and 2** of the revised paper. With these baselines included, the evaluation now spans conventional GIB-based explainers, recent GIB variants, and modern training-free methods, providing a more balanced and up-to-date comparison across different categories of graph explanation techniques.
>
> The tables below summarize the bimodality and binarization score of the two non-GIB explainers alongside our method. While certain non-GIB methods appear to achieve relatively high distribution scores on specific datasets, these cases do not consistently reflect reliable explanation, considering **Table 1** and **Table 2** in **Section 4**. In contrast, our method consistently improves both the shape of the score distribution and the correctness of the resulting explanations.
>
> | Bimodality | BA3 | Mutagenicity | MNIST | FC | Benzene |
> | -- | -- | -- | -- | -- | -- |
> | Eig-Search | 0.422 | 0.558 | 0.672 | 0.498 | 0.727 |
> | GOAt        | 0.576 | 0.636 | **0.759** | 0.759 | 0.691 |
> | 3SG-Explainer | **0.826** | **0.670** | 0.722 | **0.872** | **0.896** |
>
> | Binarization | BA3 | Mutagenicity | MNIST | FC | Benzene |
> | --- | --- | --- | --- | --- | --- |
> | Eig-Search | 0.425 | 0.531 | **0.791** | 0.456 | 0.661 |
> | GOAt        | 0.564 | 0.590 | 0.725 | 0.725 | 0.653 |
> | 3SG-Explainer | **0.835** | **0.591** | 0.666 | **0.851** | **0.890** |
>
> [1] Lu, Shengyaoe, et al. "Eig-search: Generating edge-induced subgraphs for gnn explanation in linear time." (ICML 2024)
> [2] Lu, Shengyao, et al. "GOAt: Explaining graph neural networks via graph output attribution." (ICLR 2024)
>
>
> **[W2, W4] Stability under noisy pseudo-labels**
>
> The revised paper now includes an analysis of robustness to noisy or poorly calibrated base explainers, provided in **Figure 6** and **Appendix I**. When random perturbations are injected into the pseudo-labels, the AUC remains stable across noise levels, indicating that predictive performance is not highly sensitive to such imperfections. The polarization metrics exhibit only mild variability and retain mean values comparable to those observed in the noise-free setting. This shows that the score distribution continues to move toward clearer extremes and remains sufficiently polarized even when noise is introduced. Although each round incorporates additional noise, the overall process does not accumulate or amplify noise across iterations, allowing the method to maintain stable behavior throughout the self-guided updates.
>
> Beyond these empirical results, we note that methods relying on self-generated supervisory signals cannot always guarantee perfect correction in the absence of ground-truth labels. The practical question is whether the refinement mechanism can extract useful structure from imperfect predictions without drifting over iterations.
>
> Our noise-injection study demonstrates that, as long as the base explainer provides informative signals, the refinement remains stable and continues to sharpen the explanation distribution. These observations suggest that the method can still converge to meaningful polarization when the base explainer produces noisy or weakly separable confidence scores, and therefore is likely to remain effective even with imperfect pseudo-labels.

---

> ### Author Response · Authors · 2025-11-21
>
> **[W3] Robustness to hyperparameter settings**
>
> The revised version now includes an empirical sensitivity analysis for the newly introduced hyperparameters, presented in **Figure 5** and **Appendix G**. The analysis explores the full grid of $(\alpha_0, c, w^+)$ across both training rounds.
>
> The results show that performance remains stable across a broad range of settings. Across almost all hyperparameter combinations, the performance remains higher than that of the base explainer. This shows that the setting used in the main tables is not a special one-time optimum, but rather a typical configuration that performs well among many others. Although some variability is present, the overall distribution indicates that the self-guided mechanism does not rely on fine-grained hyperparameter tuning.
>
> These findings suggest that the method is robust with respect to $\alpha_0$, $c$, and $w^+$, and that many configurations lead to consistent improvements.
>
>
> **[W5] The role of the explainer**
>
> We would like to clarify the role of explainers in GNN explanation, not only in our work, but in the field as a whole. An explainer is not designed to be interpretable in and of itself. Rather, its sole purpose is to provide interpretability for the GNN by identifying which parts of the input graph most significantly influenced the model’s decision.
>
> In this sense, the explainer serves as a tool that enhances our understanding of the GNN’s behavior, not a model that needs to be transparent in its internal mechanics. What ultimately matters is whether the explainer can generate high-quality, faithful, and useful explanations that make the GNN’s decision-making process more understandable to humans.
>
> From this perspective, applying an explainer to a GNN does not constitute circular reasoning, but is a valid and well-established strategy for gaining insight into black-box predictions.
>
>
> **[W6] Qualitative evaluation and real-world interpretability**
>
> We address the reviewer’s concerns regarding qualitative evaluation, dataset diversity, and real-world interpretability. To broaden the empirical coverage, we extend our evaluation by adding an additional real-world molecular dataset, Benzene [3], which provides a chemically meaningful setting for assessing the generalizability of our method. The corresponding results have been incorporated into **Tables 1, 2, and 3**, with further analyses presented in **Appendix G, H, I, and J**.
>
> Additional visualization experiments have been added in **Appendix J**. For clarity and generality, we include one synthetic dataset (BA-3motifs) and two real-world molecular datasets (Mutagenicity and Fluoride-Carbonyl). The new figures present subgraph-highlighting results for these datasets and include comparisons against PGExplainer, TAGE, and ReFine.
>
> Our module serves a complementary purpose to the base explainer. Base explainers often highlight only a subset of the true functional substructure, leaving portions of the rationale unassigned because many chemically valid but irrelevant edges receive similar mid-range scores. However, the partial highlight still contains a useful signal. Our skewness-based pseudo-labeling extracts this reliable signal and discards the ambiguous middle region, producing a clean set of confident edges.
>
> The guided explainer then leverages implicit structural regularities in the molecule, such as recurring local motifs or consistent connectivity around functional groups, to propagate this signal. This propagation allows the model to fill in the missing parts of the rationale. As a result, 3SG-Explainer is able to recover complete and chemically coherent substructures even when the base explainer provides only partial or noisy indications of the underlying mechanism.
>
> Together, the expanded dataset and added visualizations offer a clearer and more comprehensive assessment of both the effectiveness and the real-world interpretability of the proposed approach.
>
>
> [3] Sanchez-Lengeling, Benjamin, et al. "Evaluating attribution for graph neural networks." (NeurIPS 2020)

---

### Official Review · Reviewer_VDcY · 2025-11-01

**Soundness:** 3
**Presentation:** 3
**Contribution:** 2
**Rating:** 4
**Confidence:** 4

**Summary:**

This paper proposes 3SG-Explainer, a novel method for enhancing the interpretability of graph neural networks (GNNs). The approach leverages a self-guided semi-supervised learning framework that integrates pseudo-label generation and score distribution modeling to produce discretized explanation masks that are more interpretable to humans. Experimental results demonstrate that 3SG-Explainer consistently outperforms existing explainers across multiple datasets, achieving superior performance in both F1 and AUC metrics.

**Strengths:**

1. The paper introduces a novel self-guided semi-supervised framework that effectively combines pseudo-labeling and distribution-based thresholding to enhance interpretability in GNNs.

2. By generating discretized explanation masks, the method produces clearer and more human-understandable interpretations compared to continuous-score explainers.

3. Extensive experiments on multiple datasets show consistent performance gains in both F1 and AUC metrics, while analyses on binarization and bimodality further validate the improved discreteness of the generated masks.

**Weaknesses:**

1. Please consider including more synthetic and real-world datasets, along with additional baselines, to further validate the effectiveness and generalizability of the proposed method.
2. The introduction of the Guided Explainer module may introduce additional computational overhead. It would be helpful to provide a complexity analysis or runtime comparison to clarify its efficiency.
3. The method uses PGExplainer as the base explainer; however, PGExplainer is known to suffer from out-of-distribution (OOD) issues, which may lead to inaccurate masks and, consequently, noisy pseudo-labels. Please provide further clarification or experiments to address this concern.

**Questions:**

1. It is known that GIB-based methods such as PGExplainer are quite sensitive to hyperparameter settings. It would be helpful to include additional analysis or discussion based on their hyperparameter configurations to further explain the limitations of GIB-based methods in mask discretization capability. Moreover, approaches such as adopting a top-k selection strategy or increasing the weight of regularization losses may also lead to more discrete mask outputs. Therefore, additional clarification is needed to justify the necessity and distinct advantage of the proposed method in explicitly addressing the mask discretization problem.
2. As shown in Figure 3, except for the BA3 dataset, the mask optimization appears suboptimal in other datasets. For instance, in the Fluorid-Carbonyl dataset, some important edges are misclassified as unimportant, while in Mutagenicity, the non-important mask distribution seems scattered. Please consider adding further explanations or supplementary experiments.
3. It is unclear how the thresholds are determined and how sensitive the explainer is to these threshold values. Please include a hyperparameter sensitivity analysis to strengthen the experimental evidence.

---

> ### Author Response · Authors · 2025-11-21
>
> We appreciate the reviewer’s valuable suggestions and insightful questions. We provide detailed responses below to address each issue carefully.
>
> **[W1] Experimental results with additional baselines and datasets**
>
> In the revised paper, we additionally include two recent baselines, Eig-Search [1] and GOAt [2], both introduced in 2024 as training-free, non-GIB explainers. Incorporating these methods expands the evaluation beyond the previously considered GIB-based approaches and broadens the methodological coverage. The detailed results for these new baselines can be found in **Tables 1 and 2** of the revised paper. With these baselines included, the evaluation now spans conventional GIB-based explainers, recent GIB variants, and modern training-free methods, providing a more balanced and up-to-date comparison across different categories of graph explanation techniques.
>
> We also extend our evaluation by adding an additional real-world dataset, Benzene [3], which offers a useful setting for assessing the generalizability of the method on chemically meaningful graph structures. Experiments on the Benzene dataset have been incorporated into **Tables 1, 2, and 3**, and further qualitative and quantitative results are provided in **Appendix G, H, I, J** for completeness.
>
> The addition of the new baselines and the Benzene dataset provides a broader evaluation setting and helps more clearly demonstrate both the effectiveness and the generalizability of our method.
>
> [1] Lu, Shengyaoe, et al. "Eig-search: Generating edge-induced subgraphs for gnn explanation in linear time." (ICML 2024)
>
> [2] Lu, Shengyao, et al. "GOAt: Explaining graph neural networks via graph output attribution." (ICLR 2024)
>
> [3] Sanchez-Lengeling, Benjamin, et al. "Evaluating attribution for graph neural networks." (NeurIPS 2020)
>
> **[W2] Low computational cost of our approach**
>
> The revised version now includes an analysis of the time cost introduced by multi-round retraining, provided in **Table 7** and **Appendix H**. We report end-to-end training time including both rounds, % validation steps,
> evaluation on the validation split and the intermediate pseudo-labeling stage. Inference time is also measured on the full test split for each dataset.
>
> As shown in **Table 7**, the total training and inference cost remains competitive with, and often lower than that of several single stage explainers. The guided round with pseudo-labeling adds only a small overhead. Even on the large and dense dataset in our experiments, the overall runtime remains low, indicating that the approach scales effectively with graph size and density.
>
> These results show that the multi-round self-guided procedure does not introduce a meaningful computational burden although it is attached as as complement and maintains efficiency comparable to single stage baselines.
>
>
>
> **[W3] Response to the OOD-related question**
>
> We fully agree that OOD robustness is an important aspect of explainer reliability. To ensure that our framework is not tied to PGExplainer’s limitations, we evaluated our method using explainers explicitly developed to alleviate OOD issues in **Table 4**. Across these settings, the guided explainer continues to perform consistently well, suggesting that our approach is not reliant on any particular vulnerability of PGExplainer but instead remains stable even under explainers with improved OOD behavior.
> Because our method functions as an attachable post-hoc refinement module rather than a standalone explainer, it naturally inherits the OOD robustness of whichever base explainer it is paired with, helping to alleviate the concern raised in the comment.
>
> To further examine stability under imperfect supervisory signals, we additionally evaluate the method in settings where the base explainer may provide noisy or partially unreliable pseudo-labels, as shown in **Figure 6** of **Appendix I**. This setup also serves as a proxy for situations where the base explainer degrades due to factors such as OOD behavior.
>
> Even under injected perturbations, the AUC remains stable and the polarization metrics preserve mean values close to the noise-free case, indicating that the refinement does not drift or amplify errors across rounds. These results suggest that the method is robust to degraded pseudo-labels, making the refinement reliable across a broad range of practical scenarios.

---

> ### Author Response · Authors · 2025-11-21
>
> **[Q1] Clarification on mask discretization**
>
> GIB-based explainers are widely used because they provide a principled formulation for balancing informativeness and sparsity, which are central goals in explanation learning. However, as provided in **Section 1** and **Section 3.3**, the standard GIB objective exhibits a limitation: even with sparsity and entropy regularization, the optimization often converges to values stuck in the middle rather than producing sufficiently polarized values. This tendency arises from the fixed-point behavior of the regularized GIB formulation and makes stable discretization difficult.
>
> We additionally examined whether increasing the entropy regularization term in GIB-based explainers could improve mask discretization. As shown in the given Table A, adjusting the regularizer produces substantial variation in the distribution scores. We observed that manually amplifying the entropy regularizer does not reliably produce meaningful discretization and may even harm explanation correctness. This further highlights the need for a refinement mechanism that reshapes the score distribution in a stable and principled way rather than relying on heuristic regularization tuning.
>
> ### Table A. Effect of entropy regularization on BA-3Motifs (PGExplainer)
> | entropy regularizer | AUC (mean ± std) | Binarization (mean ± std) | Bimodality (mean ± std) |
> |---|---|---|---|
> | **0.001** | 0.746 ± 0.066 | 0.683 ± 0.321 | 0.565 ± 0.337 |
> | **0.01**  | 0.746 ± 0.066 | 0.683 ± 0.321 | 0.566 ± 0.337 |
> | **0.1**   | 0.747 ± 0.066 | 0.683 ± 0.320 | 0.569 ± 0.335 |
> | **1.0**   | 0.797 ± 0.075 | 0.693 ± 0.309 | 0.620 ± 0.331 |
> | **10.0**  | 0.681 ± 0.181 | 0.679 ± 0.266 | 0.641 ± 0.282 |
> | **100.0** | 0.629 ± 0.194 | 0.735 ± 0.250 | 0.733 ± 0.261 |
> | **1000**  | 0.627 ± 0.194 | 0.728 ± 0.255 | 0.727 ± 0.265 |
>
>
> Our approach instead focuses on reshaping the score distribution itself. By explicitly guiding the scores toward polarized modes before thresholding, important and unimportant edges become more separable, enabling natural and stable discretization without relying on manual sparsity controls.
>
> The revised manuscript includes an expanded analysis of hyperparamter robustness in **Appendix G**, showing that this behavior remains consistent across a wide range of hyperparameter configurations. The results demonstrate that the proposed mechanism is robust and does not depend on fine-tuned choices to achieve reliable polarization.
>
> **[Q2] Clarification on the suboptimality in Figure 3**
>
> The suboptimal mask shapes seen in **Figure 3** are expected, as they directly reflect the inherent noise and instability of the base explainer. Since our goal is to evaluate 3SG-Explainer under a wide variety of realistic conditions, we intentionally report results across datasets where the base explainer performs well (e.g., BA-3motifs) and where it performs suboptimally (e.g., Mutagenicity, Fluoride-Carbonyl). Thus, the figures are not meant to show that every scenario yields the best possible base mask, but rather to reveal the diverse and often noisy distributions that existing GIB-based explainers naturally produce.
>
> Importantly, our emphasis is on the relative change in distribution after applying the self-guided refinement. Even when the base scores are noisy or scattered, 3SG-Explainer consistently sharpens their shapes, yielding clearer polarization and improved separation between important and unimportant edges across rounds. This relative improvement is exactly what **Figure 3** is intended to highlight rather than the absolute optimality of the base mask.
>
>
> **[Q3] Robustness to hyperparameter settings**
>
> The revised version now includes an empirical sensitivity analysis for the newly introduced hyperparameters, presented in **Figure 5** and **Appendix G**. The analysis explores the full grid of $(\alpha_0, c, w^+)$ across both training rounds.
>
> The results show that performance remains stable across a broad range of settings. Across almost all hyperparameter combinations, the performance remains higher than that of the base explainer. This shows that the setting used in the main tables is not a special one-time optimum, but rather a typical configuration that performs well among many others. Although some variability is present, the overall distribution indicates that the self-guided mechanism does not rely on fine-grained hyperparameter tuning.
>
> These findings suggest that the method is robust with respect to $\alpha_0$, $c$, and $w^+$, and that many configurations lead to consistent improvements.

---

### Official Review · Reviewer_xhoi · 2025-11-01

**Soundness:** 2
**Presentation:** 3
**Contribution:** 3
**Rating:** 4
**Confidence:** 4

**Summary:**

This paper introduces 3SG-Explainer, a self-guided multi-stage framework for explaining GNNs. It transforms soft edge-importance scores from a base explainer into pseudo-labels and trains a guided GNN explainer under direct supervision. A label-conditioned graph generator augments pseudo-labeled graphs to enhance robustness. Theoretical analysis suggests a tighter VC-dimension–based generalization bound than single-stage explainers. Experiments on four benchmarks show consistent AUC gains and stable performance across various base explainers and generators.

**Strengths:**

- Clear motivation and simple formulation. The paper clearly identifies the limitation of non-binarized explanations and proposes a clean, self-guided remedy without architectural complexity.

- Sound theoretical grounding. The analysis of GIB’s non-binarization, BCE-driven polarization, and tail-risk generalization offers valuable intuition and rigor.

- Empirical validation. Improvements in both performance (AUC/F1) and interpretability metrics (binarization, bimodality) are consistent across datasets and base explainers.

- Readable and modular. The framework can be readily integrated with existing explainers and GNN architectures.

**Weaknesses:**

- Dependence on pseudo-label quality. The method assumes reliable confidence estimation from the base explainer; if those scores are noisy, pseudo-label errors may propagate. While the theory includes a tail-noise term, empirical robustness tests are limited.

- Hyperparameter sensitivity. The approach introduces new hyper-parameters: confidence-quantile threshold alpha_0,  skew-adjustment factor c, and a class weight for balancing positive edges. However, the paper does not provide a detailed analysis of how these parameters affect performance or stability. This lack of sensitivity study partially offsets the simplicity of the design and makes it unclear how robust the method is to different hyperparameter settings.

- Efficiency not reported. Multi-round retraining increases computational cost, yet runtime or scaling comparisons with single-round baselines are not provided.

- Limited qualitative insight. Although the distributional metrics show better polarization, there is little qualitative or domain-specific evidence that the resulting subgraphs are semantically meaningful.

**Questions:**

- How robust is the method to noisy or poorly calibrated base explainers? For instance, if the confidence scores are less separable, does the self-guided training still converge to meaningful polarization?

- Can you analyze how the newly introduced hyper-parameter affect the performance?  A empirical sensitivity analysis is needed here. Is there a practical guideline or heuristic for tuning them across datasets?

- What is the additional training-time cost introduced by multi-round retraining, and how does it scale with graph size or number of rounds compared to single-stage explainers?

- Can you provide more qualitative examples or case studies (e.g., molecular motifs) demonstrating that the polarized explanations correspond to meaningful structural patterns?

---

> ### Author Response · Authors · 2025-11-21
>
> We sincerely appreciate the reviewer’s careful and detailed comments. We have provided specific responses to each concern and hope that our responses provide clear and satisfactory clarification.
>
> **[W1, Q1] Stability under noisy pseudo-labels**
>
> The revised paper now includes an analysis of robustness to noisy or poorly calibrated base explainers, provided in **Figure 6** of **Appendix I**. When random perturbations are injected into the pseudo-labels, the AUC remains stable across noise levels, indicating that predictive performance is not highly sensitive to such imperfections.
> The polarization metrics exhibit only mild variability and retain mean values comparable to those observed in the noise-free setting. This shows that the score distribution continues to move toward clearer extremes and remains sufficiently polarized even when noise is introduced.
>
> These observations suggest that the method can still converge to meaningful polarization when the base explainer produces noisy or weakly separable confidence scores, and therefore is likely to remain effective even with imperfect pseudo-labels.
>
> **[W2, Q2] Robustness to hyperparameter settings**
>
> The revised version now includes an empirical sensitivity analysis for the newly introduced hyperparameters, presented in **Figure 5** and **Appendix G**. The analysis explores the full grid of $(\alpha_0, c, w^+)$ across both training rounds.
>
> The results show that performance remains stable across a broad range of settings. Across almost all hyperparameter combinations, the performance remains higher than that of the base explainer. This shows that the setting used in the main tables is not a special one-time optimum, but rather a typical configuration that performs well among many others. Although some variability is present, the overall distribution indicates that the self-guided mechanism does not rely on fine-grained hyperparameter tuning.
>
> These findings suggest that the method is robust with respect to $\alpha_0$, $c$, and $w^+$, and that many configurations lead to consistent improvements.
>
> **[W3, Q3] Low computational cost of our approach**
>
> The revised version now includes an analysis of the time cost introduced by multi-round retraining, provided in **Table 7** and **Appendix H**. We report end-to-end training time including both rounds, % validation steps,
> \blue{evaluation on the validation split} and the intermediate pseudo-labeling stage. Inference time is also measured on the full test split for each dataset.
>
> As shown in Table 7, the total training and inference cost remains competitive with, and often lower than that of several single stage explainers. The guided round with pseudo-labeling adds only a small overhead. Even on the large and dense dataset in our experiments, the overall runtime remains low, indicating that the approach scales effectively with graph size and density.
>
> These results show that the multi-round self-guided procedure does not introduce a meaningful computational burden although it is attached as as complement and maintains efficiency comparable to single stage baselines.
>
> **[W4, Q4] Additional qualitative comparisons**
>
> Additional visualization experiments have been added in **Appendix J**. For clarity and generality, we include one synthetic dataset (BA-3motifs) and two real-world molecular datasets (Mutagenicity and Fluoride-Carbonyl). The new figures present subgraph-highlighting results for these datasets and include comparisons against PGExplainer, TAGE, and ReFine.
>
> Our module serves a complementary purpose to the base explainer. Base explainer often highlights only a subset of the true functional substructure, leaving portions of the rationale unassigned because many chemically valid but irrelevant edges receive similar mid-range scores. However, the partial highlight stilly contains a useful signal. Our skewness-based pseudo-labeling extracts this reliable signal and discards the ambiguous middle region, producing a clean set of confident edges.
>
> The guided explainer then leverages implicit structural regularities in the molecule, such as recurring local motifs or consistent connectivity around functional groups, to propagate this signal. This propagation allows the model to fill in the missing parts of the rationale. As a result, 3SG-Explainer is able to recover complete and chemically coherent substructures even when the base explainer provides only partial or noisy indications of the underlying mechanism.

---

### Official Review · Reviewer_yC86 · 2025-11-01

**Soundness:** 3
**Presentation:** 3
**Contribution:** 3
**Rating:** 4
**Confidence:** 4

**Summary:**

This paper proposes 3SG-Explainer, which generates clear explanatory subgraphs by polarizing edge-importance scores. 3SG-Explainer first employs a pretrained base explainer to produce soft importance scores, then uses skewness-adaptive analysis to identify high-confidence regions and assign pseudo-labels to them. Finally, a guided explainer is trained in a semi-supervised manner, where the GNN architecture propagates supervision signals to unlabeled edges. Extensive experiments on four benchmark datasets demonstrate the effectiveness of the proposed approach.

**Strengths:**

1. This paper provides a rigorous theoretical foundation for deriving the binarization for GIB.
2. The paper is easy to follow, and the figures are well-designed and visually appealing.
3. Extensive experiments of this paper support the method proposed in this paper.

**Weaknesses:**

1. The construction of the proposed method lacks a clear motivation. For example, in Equation 1, why was Fisher’s moment coefficient of skewness chosen to measure skewness?
2. Could the authors provide more visualization experiments to intuitively demonstrate the model’s effectiveness?
3. The paper could include some more recent baselines to strengthen the persuasiveness of the results, such as RegExplainer.
4. In Figure 4, why does the F1 score for some dataset remain almost unchanged, even though other metrics (bimodality and binarization score) continue to improve in later rounds?

**Questions:**

Please see weaknesses.

---

> ### Author Response · Authors · 2025-11-21
>
> We sincerely thank the reviewer for the helpful and thoughtful comments. Below, we respond to each point and hope our answers address the concerns raised.
>
> **[W1] Clarifying the motivation behind 3SG-Explainer**
>
> - General motivation: A core motivation of our method is that existing GNN explainers, particularly GIB-based approaches, consistently output continuous edge scores that concentrate around the mid-range rather than forming clear binary-like decisions. We start from the fact that we do not know which edges are truly important in advance and must rely on the score distribution produced by the base explainer. With a skewness-based criterion, we identify the reliable edges and generate binary pseudo-labels. To make effective use of this incomplete supervision, we adopt a semi-supervised learning scheme that can exploit the confident labels. As the guided explainer, we employ a lightweight GNN classifier because it provides fast inference and matches the structure of the refinement task.
>
> - Fisher’s moment coefficient: We selected Fisher’s moment coefficient because our goal is to capture tail asymmetry in the score distribution. Fisher skewness is a standard, closed-form, and differentiable statistic that reflects third-order moment imbalance and is highly sensitive to heavy tails-a common characteristic of GIB-based explainers.
> In addition, moment-based measures offer a principled way to summarize the global shape of the distribution, making Fisher’s coefficient a particularly coherent choice for determining asymmetric cutoffs. It provides a stable signal that allows us to avoid noisy mid-range regions.
>
> **[W2] Additional qualitative comparisons**
>
> Additional visualization experiments have been added in **Appendix J**. For clarity and generality, we include one synthetic dataset (BA-3motifs) and two real-world molecular datasets (Mutagenicity and Fluoride-Carbonyl). The new figures present subgraph-highlighting results for these datasets and include comparisons against PGExplainer, TAGE, and ReFine.
>
> Base explainers often highlight only a subset of the true functional substructure, leaving portions of the motifs unassigned.
> However, our skewness-based pseudo-labeling extracts meaningful signal from base explainers, producing a clean set of confident edges while discarding the ambiguous region. The guided explainer then leverages implicit structural regularities, such as local patterns or consistent connectivity around functional groups, to propagate this signal. This propagation allows the model to fill in the missing parts of the rationale. As a result, 3SG-Explainer is able to recover complete substructures even when the base explainer provides partial or noisy indications.
>
>
> **[W3] Experimental results with additional baselines**
>
> In the revised paper, we additionally include two recent baselines, Eig-Search [1] and GOAt [2], both introduced in 2024 as training-free, non-GIB explainers. Incorporating these methods expands the evaluation beyond the previously considered GIB-based approaches and broadens the methodological coverage. The detailed results for these new baselines can be found in **Tables 1 and 2** of the revised paper.
> With these baselines included, the evaluation now spans conventional GIB-based explainers, recent GIB variants, and modern training-free methods, providing a more balanced and up-to-date comparison across different categories of graph explanation techniques.
>
> [1] Lu, Shengyaoe, et al. "Eig-search: Generating edge-induced subgraphs for gnn explanation in linear time." (ICML 2024)
>
> [2] Lu, Shengyao, et al. "GOAt: Explaining graph neural networks via graph output attribution." (ICLR 2024)
>
>
> **[W4] Explanation for F1 saturation in Figure 4**
>
> We design our self-guidance procedure to explicitly sharpen the explanation score distribution toward a bimodal, binary form while maintaining the high F1 score.
> Since the bimodality and binarization measures are aligned with this objective, they naturally continue to improve as we add rounds and further polarize edge scores.
> In contrast, F1 depends only on thresholded rankings, so once this ranking stabilizes in the early rounds, additional sharpening has little effect on F1; empirically, later rounds do not hurt the F1 score and usually preserve or slightly improve it while clearly enhancing interpretability.

---

### Author Response · Authors · 2025-11-21

We thank the reviewers for their constructive and detailed feedback.
In response, we have revised both the main paper and the appendix to strengthen empirical validation and expand qualitative and quantitative analysis.
Below, we summarize the major changes made in the revised manuscript.

## **1. Additional Baselines**
We incorporated two recent non-GIB explainers, **Eig-Search** and **GOAt**, into the main quantitative evaluation.
These additions appear in **Tables 1 and 2**, broadening coverage beyond the originally included GIB-based methods.
Across datasets, our method continues to outperform these baselines by a clear margin.


## **2. Additional Real-World Dataset**
We added the **Benzene** molecular dataset to improve real-world evaluation and generalizability.
Experimental results for this dataset were integrated into **Tables 1, 2, and 3**, with additional discussion in **Appendix G-I**.
On this dataset as well, the our method achieves strong performance and maintains consistent gains over baselines.

## **3. Robustness Analysis**
We introduced two complementary robustness studies:
- Hyperparameter sensitivity analysis (**Appendix G, Figure 5**)
- Noise-injection robustness (**Appendix I, Figure 6**)

These experiments show that performance remains stable under a wide range of settings and imperfect pseudo-labels.

## **4. Time-Cost and Efficiency Analysis**
We added a detailed analysis of computational cost, including both training-time overhead and inference efficiency.
The full results are presented in **Appendix H** and **Table 7**, clarifying that the multi-round refinement introduces only a small overhead relative to the base explainer.

## **5. Expanded Visualization Results**
We added extensive qualitative visualizations for BA-3motifs, Mutagenicity, and Fluoride-Carbonyl with comparisons to various baselines.
These results, included in **Appendix J**, illustrate that the refinement step recovers more complete and coherent rationale subgraphs.

---

### Author Response · Authors · 2025-11-27

Dear Reviewers,


Thank you for your thoughtful and constructive feedback. We have carefully addressed each of your comments and provided detailed responses below. Please feel free to inform us if there are any such concerns.


Warm regards,

Authors

---

### Author Response · Authors · 2025-12-02
**Summary Comment for Area Chairs**

Dear Area Chairs,

Due to the unexpected disruption in this year’s review process, the reviewers have not been able to engage in the discussion phase. Given this situation, we would like to provide a concise summary of our revisions and clarifications so that the ACs can effectively assess the updated manuscript.
Below, we highlight the major contributions and concerns mentioned by the reviewers and summarize the key changes implemented in the revised version of the paper.

We are encouraged that the reviewers recognized the contributions and significance of our work, particularly highlighting the following strengths:

- **Theoretical rigor and clarity (yC86, xhoi, KmWJ):**
  Reviewers noted that the paper provides a rigorous and well-grounded theoretical foundation for understanding why GIB-based methods fail to binarize and how the proposed semi-supervised BCE objective leads to polarized distributions. They found the analysis intuitive, valuable, and conceptually strengthening.

- **Clear motivation and well-structured formulation (xhoi, KmWJ):**
  The motivation for addressing non-binarized explanations was viewed as clear and compelling. Reviewers appreciated that the proposed self-guided refinement is simple, principled, and free of unnecessary architectural complexity.

- **Empirical strength and interpretability improvements (yC86, xhoi):**
  Reviewers highlighted that the experiments are extensive, solid, and consistently demonstrate improvements in both correctness (AUC/F1) and interpretability metrics (binarization, bimodality) across datasets and explainers.

- **Practical applicability and modular design (xhoi):**
  The framework was noted to be readable, easy to follow, and readily compatible with existing explainers and GNN architectures.

We carefully considered the reviewers’ concerns and addressed each of them in our rebuttal with additional experiments, and clarifications. Specifically, we responded by including the following:

- **Expanded baselines (yC86, VDcY, KmWJ):**
    We added two non-GIB explainers, **Eig-Search** and **GOAt**, to broaden methodological coverage. These appear in **Tables 1–2**, and our method retains clear improvements over them.

- **Additional real-world dataset (VDcY, KmWJ):**
    We incorporated the **Benzene** molecular dataset to strengthen generalizability. Results are included in **Tables 1–3** and discussed in **Appendix G–I**.

- **Robustness analysis (xhoi, VDcY, KmWJ):**
    We added hyperparameter sensitivity analysis (**Appendix G, Fig. 5**)  and noise-injection robustness study (**Appendix I, Fig. 6**) Both show that the refinement remains stable under wide hyperparameter ranges and imperfect pseudo-labels.

- **Computational efficiency (xhoi, VDcY):**
    We included a time-cost analysis (**Appendix H, Table 7**) evaluating both training overhead and inference speed.
    Results show that the additional refinement round adds only minor overhead relative to base explainers.

- **Expanded qualitative evaluation (yC86, xhoi, KmWJ):**
    We added extensive visualizations on BA-3motifs, Mutagenicity, and Fluoride–Carbonyl (**Appendix J**), demonstrating that our refinement consistently reconstructs more complete and coherent rationale subgraphs than baselines.

We are very grateful to the ACs for their time and effort in evaluating our paper, and we hope that this summary assists in efficiently assessing the updated submission.

Thank you again for your valuable support.

Best regards,
Authors

---

### Meta-Review · Area_Chair_MmFC · 2026-01-06

**Summary:**

This paper proposes 3SG-Explainer to address the issue of non-binarized edge importance scores produced by existing explainers. The paper is clearly motivated and presented. The experiments are conducted across multiple datasets and compatible with a wide range of base explainers.

The reviewers mainly raised concerns about the motivation and design choices, visualization experiments, additional baselines and datasets, the robustness of pseudo-labeling, hyperparameter sensitivity, computational efficiency, and potential OOD issues. While the authors made a strong effort to address the reviewers’ comments by adding experiments, analyses, and clarifications, several concerns are still not fully addressed.

Considering the novelty of the proposed approach as well as the extent of the experimental evaluation, the paper remains slightly below the acceptance threshold.

**Reviewer Concerns:**

Reviewer yC86 questioned the motivation for using Fisher’s skewness, more visualization experiments, additional baselines, and the saturation of the F1 score. Reviewer xhoi raised concerns about pseudo-label noise and robustness, hyperparameter sensitivity, computational efficiency, and visualization results. Reviewer VDcY pointed out limitations in datasets and baselines, computational efficiency, and potential OOD issues induced by PGExplainer. Reviewer KmWJ also noted concerns about adding more baselines, pseudo-label noise and robustness, hyperparameter sensitivity and visualization experiments.

The authors conducted many experiments to address them, including adding two baselines Eig-Search and GOAt, incorporating the Benzene molecular dataset, adding hyperparameter sensitivity, adding pseudo-label noise, analyzing computational efficiency and visualization results.

While the authors made a strong effort to address the reviewers’ comments by adding experiments, analyses, and clarifications, several concerns remain. For pseudo-label noise, the maximum noise ratio is 0.1 and does not try real noise distributions encountered in real-world settings, making it unclear how noise truly affects the explanation results. Regarding OOD issues, although the authors changed the base explainer, they did not evaluate the method on OOD datasets, and thus the performance of the proposed method is not demonstrated. In addition, the mentioned method RegExplainer is neither discussed nor evaluated in the rebuttal.

**Reviewer Scores:**

Reviewer yC86 would likely slightly increase their score, Reviewer xhoi might slightly increase or keep the score unchanged, while Reviewers VDcY and KmWJ would most likely keep their original scores.

---

### Decision · Program_Chairs · 2026-01-26

Reject